# Cryo electron tomography with volta phase plate reveals novel structural foundations of the 96-nm axonemal repeat in the pathogen *Trypanosoma brucei*

Simon Imhof[1†], Jiayan Zhang[1,2,3†], Hui Wang[1,3,4], Khanh Huy Bui[5], Hoangkim Nguyen[1‡], Ivo Atanasov[3], Wong H Hui[3], Shun Kai Yang[5], Z Hong Zhou[1,2,3,4*], Kent L Hill[1,2,3*]

[1]Department of Microbiology, Immunology and Molecular Genetics, University of California, Los Angeles, Los Angeles, United States; [2]Molecular Biology Institute, University of California, Los Angeles, Los Angeles, United States; [3]California NanoSystems Institute, University of California, Los Angeles, Los Angeles, United States; [4]Department of Bioengineering, University of California, Los Angeles, Los Angeles, United States; [5]Department of Anatomy and Cell Biology, McGill University, Montreal, United States

*For correspondence:
Hong.Zhou@UCLA.edu (ZHZ);
kenthill@microbio.ucla.edu (KLH)

†These authors contributed equally to this work

Present address: ‡Teva Pharmaceuticals, California, United States

Competing interests: The authors declare that no competing interests exist.

**Abstract** The 96-nm axonemal repeat includes dynein motors and accessory structures as the foundation for motility of eukaryotic flagella and cilia. However, high-resolution 3D axoneme structures are unavailable for organisms among the Excavates, which include pathogens of medical and economic importance. Here we report cryo electron tomography structures of the 96-nm repeat from *Trypanosoma brucei*, a protozoan parasite in the Excavate lineage that causes African trypanosomiasis. We examined bloodstream and procyclic life cycle stages, and a knockdown lacking DRC11/CMF22 of the nexin dynein regulatory complex (NDRC). Sub-tomogram averaging yields a resolution of 21.8 Å for the 96-nm repeat. We discovered several lineage-specific structures, including novel inter-doublet linkages and microtubule inner proteins (MIPs). We establish that DRC11/CMF22 is required for the NDRC proximal lobe that binds the adjacent doublet microtubule. We propose that lineage-specific elaboration of axoneme structure in *T. brucei* reflects adaptations to support unique motility needs in diverse host environments.

## Introduction

Flagella (also called cilia) are hair-like structures that protrude from the surface of eukaryotic cells and perform motility and signaling functions (*Smith and Rohatgi, 2010*). These activities are essential for health, development and reproduction in humans and other multicellular organisms and to power movement of protists, including microbial pathogens that afflict nearly one billion people worldwide and present an economic burden as agricultural pests (*Langousis and Hill, 2014*; *Gerdes et al., 2009*; *Ibanez-Tallon, 2003*; *Anvarian et al., 2019*).

The structural basis for the flagellum is the axoneme, and in motile flagella the axoneme typically has a '9+2' arrangement, consisting of 9 doublet microtubules (DMTs) arrayed symmetrically around a pair of singlet microtubules, with radial spokes (RS) extending inward from each DMT contacting the central pair (*Khan and Scholey, 2018*). Axoneme beating is driven by dynein motors and associated structures arranged in a repeating unit of 96-nm periodicity along each DMT. This 96-nm

**eLife digest** The parasites that cause African sleeping sickness, known as trypanosomes, propel themselves forward using a structure called a flagellum, a bit like the tail of a human sperm. But rather than connect to the body of the cell just at the base, like in a sperm, the parasite flagellum runs along the side of the cell. This means that, when it beats, the whole cell twists in a screw-like motion. The parasite flagellum beats vigorously, changes direction often, and puts the cell under lots of mechanical stress. This unusual motion likely helps the parasites to move through a thick and sticky fluid like blood.

The similarities between the parasite flagellum and the flagellum on a human sperm are down to a shared evolutionary history. Both structures contain the same basic molecular skeleton, known as the axoneme. The axoneme contains a combination of supporting proteins and molecular motors, and the molecular motors essentially pull on the supports to bend the flagellum.

The unusual movement of trypanosome parasites suggests that their axonemes may have unique structural features. But the three-dimensional structure of trypanosome axonemes had previously not been studied in great detail. Imhof, Zhang et al. now address this gap in knowledge using a technique called "cryo electron tomography" and showed that axoneme structure in trypanosomes does share many features with those of other organisms but it has extra proteins and connections for support, which could help to protect the flagellum from mechanical stress.

The similarities and differences between human and trypanosome flagella could indicate new drug targets that could be used to protect us against these parasites. A better understanding of how flagella work in general could also give insights into human genetic diseases that involve problems with these structures.

axonemal repeat is thus the foundational unit of motility for eukaryotic flagella. Canonical features of the repeat are four outer arm dyneins (OAD) (each having two or three motor domains, depending on species), seven inner arm dyneins (IAD) (one, IAD-f, having two motor domains and the others having a single motor domain), the nexin dynein regulatory complex (NDRC) inter-doublet linkage, and two or three RS (*Porter and Sale, 2000*). The most proximal IAD in the 96nm repeat, IAD-f, is distinguished from other IADs by having two motor domains, a large Intermediate Chain/Light Chain (IC/LC) complex that connects to the OAD and the NDRC, and extra connections to the A-tubule (*Nicastro et al., 2006*; *Heuser et al., 2012a*). Within each 96-nm repeat, dynein motors are permanently affixed to the A-tubule of one DMT and use ATP-dependent binding, translocation and release of the B-tubule on the adjacent DMT to drive microtubule sliding (*Gibbons and Rowe, 1965*). DMT attachment to the basal body at one end, together with ATP-independent connections, called nexin links, between adjacent DMTs, limits sliding and therefore causes DMTs to bend in response to dynein activity (*Satir, 1968*; *Satir et al., 2014*; *Holwill and Satir, 1990*). Precise, spatio-temporal coordination of dynein activity on different DMTs enables the bend to be propagated along the length of the axoneme, giving rise to axonemal beating (*Satir, 1968*; *Lin and Nicastro, 2018*). RS, together with the NDRC and the IAD-f-IC/LC complex, are thought to provide a means for transmitting mechanochemical signals across the axoneme as part of a complex and as yet incompletely understood system for regulating dynein activity (*Porter and Sale, 2000*; *Satir et al., 2014*; *King, 2018*; *Viswanadha et al., 2017*).

Recent advances in cryo electron tomography (cryoET) have made high-resolution, 3D structural analyses of the 96-nm repeat possible, providing insights into mechanisms of axoneme assembly and motility (*Nicastro et al., 2006*; *Lin and Nicastro, 2018*; *Bui et al., 2009*; *Oda et al., 2014a*; *Jordan et al., 2018*). However, such analyses have been limited to a restricted number of cell types and phylogenetic lineages. In particular, there has been no such analysis of the 96-nm repeat in any member of the Excavates (*Figure 1*), which includes several human and agricultural pathogens of importance to global public health. Consequently, we lack understanding of the full range of structural foundations for axoneme assembly and motility, and what structural variations underlie lineage-specific beating patterns observed in different organisms. For pathogens, such variations present potential therapeutic targets.

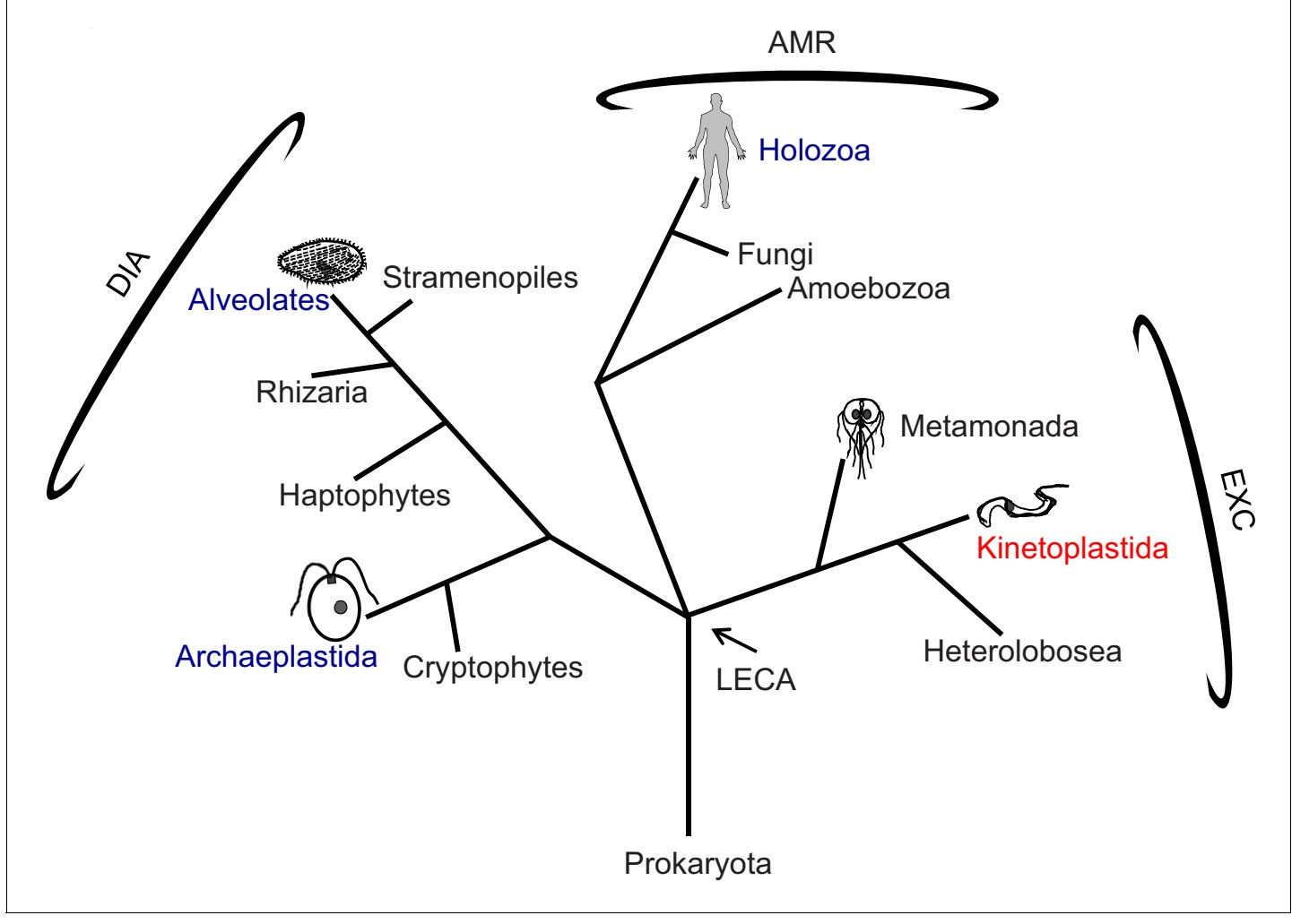

**Figure 1.** Phylogenetic tree of eukaryotes. The tree is adapted from *Dacks and Field (2018)* and *Adl et al. (2019)*. High-resolution structures of the 96-nm repeat of the axoneme are published for the clades indicated in blue, with the corresponding organism depicted in cartoon. *T. brucei* is in the clade Kinetoplastida, indicated in red, and represents the Excavates (EXC) that includes other pathogens, such as *Giardia* within Metamonada, also depicted in cartoon. The position of the last eukaryotic common ancestor (LECA) is indicated. AMR: Amorphea; DIA: Diaphoretickes; and EXC: Excavates are indicated.

African trypanosomes, *Trypanosoma brucei (T. brucei)* and related species, are parasitic protists in the Euglenozoa branch of the Excavates (*Figure 1*) (*Koonin, 2010*). They are medically and economically important pathogens of humans and other mammals (*Langousis and Hill, 2014*). Critical to *T. brucei* infection of a mammalian host (*Shimogawa et al., 2018*) and to their transmission via a tsetse fly vector (*Rotureau et al., 2014*), is motility of these parasites within and through host tissues. Motility of trypanosomes is driven by a single flagellum that is laterally connected to the cell body along most of its length (*Figure 2A*) (*Langousis and Hill, 2014*; *Heddergott et al., 2012*). The *T. brucei* flagellum consists of a 9+2 axoneme and a lineage-specific extra-axonemal structure, termed the paraflagellar rod (PFR), which runs alongside the axoneme for most of its length (*Langousis and Hill, 2014*; *Hughes et al., 2012*; *Koyfman et al., 2011*; *Cachon et al., 1988*). While the PFR exerts influence on the axoneme (*Koyfman et al., 2011*; *Santrich et al., 1997*), motility itself is driven by axoneme beating, which is transmitted directly to the cell, deforming the cell membrane and underlying cytoskeleton as the waveform propagates along the axoneme (*Sun et al., 2018*). Unlike most organisms, trypanosome axoneme beating propagates from the distal tip to proximal end in a helical wave, creating torsional strain and causing the cell to rotate on its long axis as it translocates with the flagellum tip leading (*Heddergott et al., 2012*; *Walker, 1961*;

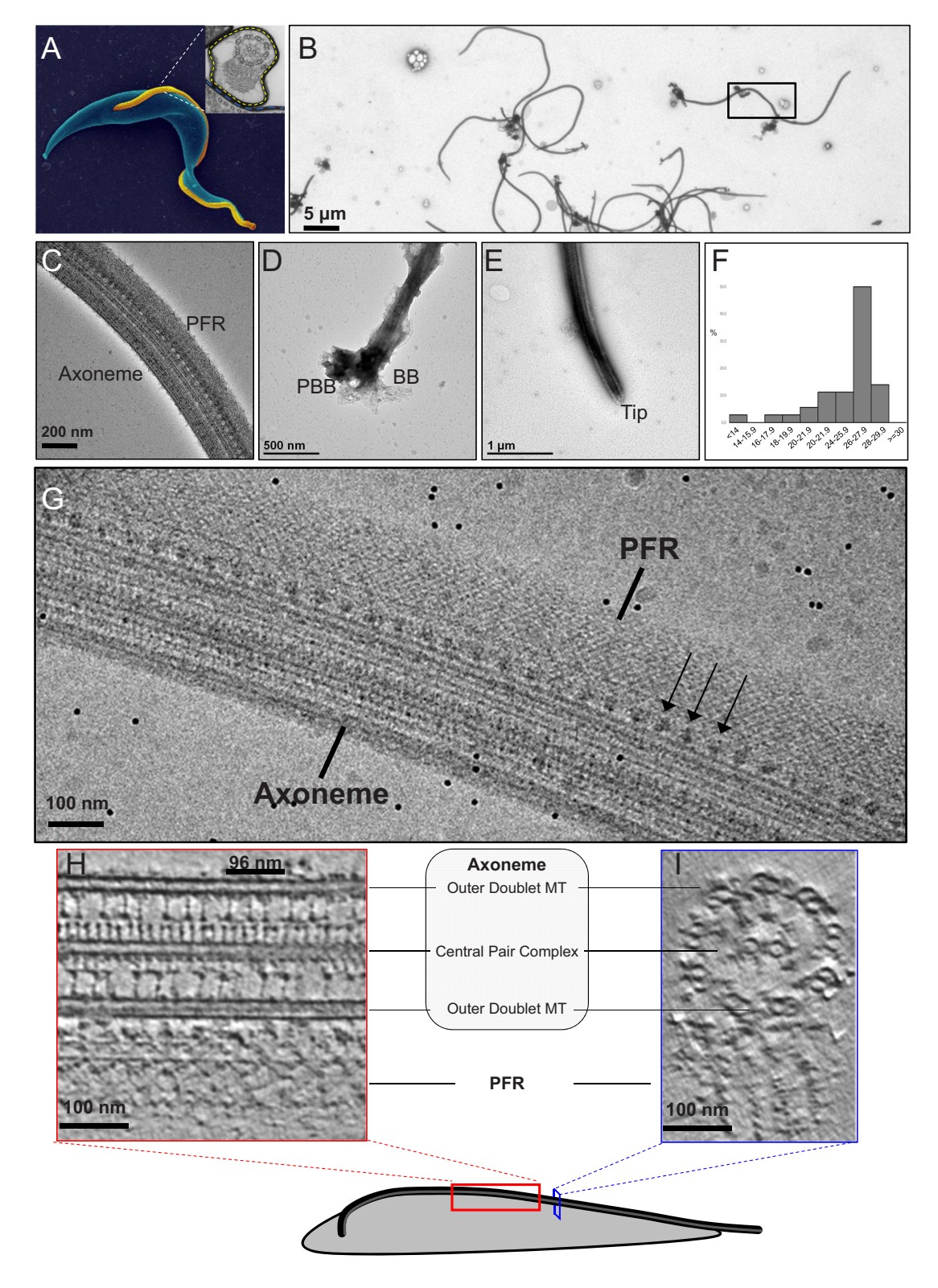

**Figure 2.** Intact demembranated flagella from BSF *T. brucei*. (**A**) A representative scanning electron microscope image of a procyclic form *T. brucei* parasite , with the cell body colored blue. The inset is a transmission electron microscope image of the flagellum from BSF *T. brucei* in representative transverse section, viewed from the proximal end, showing the 9+2 axoneme and PFR, enclosed within the flagellar membrane which is outlined by the yellow dotted line (adapted from *Hill, 2003*). (**B–E**) Negative stain TEM images of purified flagellum samples from BSF *T. brucei*, distributed on the grid

*Figure 2 continued on next page*

*Figure 2 continued*

with minimal clustering (**B**), showing that the axoneme and PFR are intact (**C**), with the basal body and pro-basal body on the proximal end (**D**), and a tapered tip at the distal end (**E**). The black box in (**B**) shows the approximate region chosen to image for cryoET. (**F**) Histogram of the length distribution of purified flagellum samples showing that the majority are full-length with a mean length of 25.2 microns (standard deviation = 3.5 microns). (**G**) A zero-degree tilted cryoEM image shows intact Axoneme, PFR and Ax-PFR connectors (arrows) from BSF *T. brucei*. (**H–I**) 6-A thick digital slice from a representative tomogram showing the sample in longitudinal (**H**) and the transverse (**I**) sections, with main structures labelled. Black line indicates one 96-nm axonemal repeat.

The online version of this article includes the following source data for figure 2:

**Source data 1.** Data for measurement of axoneme lengths in *Figure 2*, panel F.

---

*Walker and Walker, 1963*; *Rodríguez et al., 2009*) (*Videos 1* and *2*). In essence, the entire cell rotates like an auger as it moves forward. This distinctive form of locomotion provides advantages for moving in viscous environments (*Jahn and Bovee, 1968*; *Bargul et al., 2016*) such as within human and fly tissues, and gives the genus its name, as *Trypanosoma* combines the Greek words for auger (trypanon) and body (soma) (*Gruby, 1843*).

The combination of unusual locomotion mechanism, unique connections to other structures, and adaptation to diverse environmental conditions, suggests that the 96-nm repeating unit of the trypanosome axoneme might harbor lineage-specific elaborations. To investigate this possibility, we employed cryoET and sub-tomogram averaging to determine the 3D structure of the *T. brucei* 96-nm axonemal repeat. We report the 96-nm axonemal repeat structure for wild type parasites in bloodstream (BSF) and procyclic (PCF) stages, and for an RNAi knockdown targeting the CMF22/DRC11 subunit of the NDRC. Our results reveal lineage-specific adaptations, including novel inter-doublet linkages and microtubule inner proteins (MIPs). We also identify an NDRC subunit involved in inter-doublet connections between adjacent DMTs. We propose that lineage-specific adaptations to the 96-nm repeat may support the unique motility needs of these pathogens.

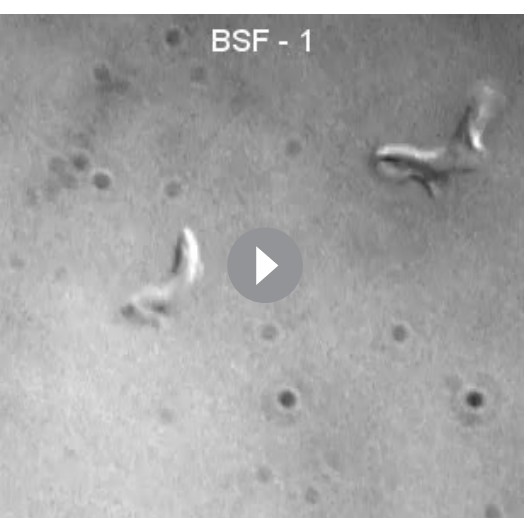

**Video 1.** Real-time video showing two *T. brucei* BSF parasites in culture medium. The two parasites collide, illustrating the need for trypanosomes to accommodate interactions with other cells and tissues, which is common in the native environment of the mammalian host and insect vector.
https://elifesciences.org/articles/52058#video1

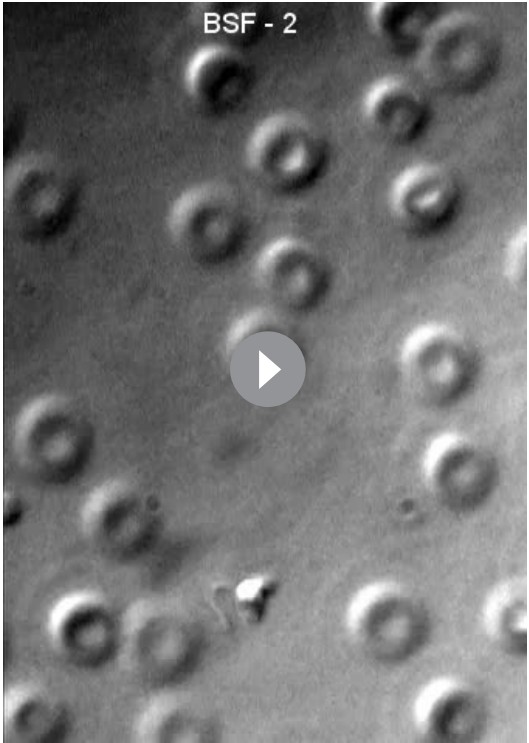

**Video 2.** Real-time video showing a *T. brucei* BSF parasites moving in mouse blood, diluted 1:100 with culture medium. Movement with flagellum tip leading and contact with host red blood cells is evident.
https://elifesciences.org/articles/52058#video2

---

## Results

### 3D structure of the trypanosome 96-nm axonemal repeat

A critical element of defining any structure is to ensure the sample is pristine. Our analyses demonstrated that flagellar skeletons purified from bloodstream form (BSF) trypanosomes are intact, including intact PFR, basal body and distal tip with uniform length distribution and a mean length of 25.2, + /- 3.5 μm (*Figure 2B–F*). Next it is critical that freezing does not distort the sample. A single zero-degree tilt image of a flagellum embedded in ice demonstrated that the axoneme, PFR and axoneme-PFR connectors remain intact following plunge freezing (*Figure 2G*). Having established high quality of vitrified samples, tilt series were collected from the center part of full-length flagella, spanning the middle third between the basal body and tip (*Figure 2B*). Major axonemal and PFR structures were resolved in slices through a single tomogram (*Figure 2H,I*, *Video 3*), indicating the 3D structure is well-preserved and relatively uncompressed (*Figure 3—figure supplement 1*).

Sub-volumes, that is particles, encompassing the 96-nm repeat of DMTs were extracted from 10 tomograms and averaged as described in Materials and methods. In total, 763 particles were averaged to determine the 3D structure of the axonemal repeat (*Figure 3A–D*, *Video 4*). The average resolution of the entire structure is 21.8 Å based on the 0.143 Fourier shell correlation criterion (*Figure 3—figure supplement 2A*). The resolutions at different regions vary based on visual inspection, and assessments by both local Fourier shell correlation (FSC) and *ResMap* (*Kucukelbir et al., 2014*) calculations (*Figure 3—figure supplement 2A,C–F*); the resolution of DMT region with MIPs reached 19.0 Å based on local FSC calculation (*Figure 3—figure supplement 2A*).

The 3D structure of the 96-nm repeat clearly resolved the expected major substructures, including OAD, IAD, RS, the IC/LC complex of IAD-f and the NDRC (*Figure 3B–E*). Individual protofilaments are well-resolved and even alpha and beta tubulin monomers within protofilaments are clearly resolved (*Figure 3F*). Several MIPs are also observed (*Figure 3B*). At this resolution, we observed a filamentous structure on the outside of the DMT that spans the entire 96-nm repeat (*Figure 3E–G*, red and white arrows). The location and extended conformation of this structure lead us to propose it to be the FAP59/172 molecular ruler described in *Chlamydomonas* that defines the 96-nm repeat (*Oda et al., 2014a*). Supporting this idea, the structure makes direct contact with RS, whose position depends on the FAP59/172 ruler (*Oda et al., 2014a*). The position of this ruler was previously determined in *Chlamydomonas* through mass-tagging, but the structure itself was not resolved (*Oda et al., 2014a*). We also observed a novel globular structure outside the B-tubule, between protofilaments B7 and B8, having a periodicity of 8 nm (*Figure 3—figure supplement 3A,B* blue arrow). The function of this structure is unknown, but it might influence dynein binding, because the microtubule binding domain of OADα contacts the B-tubule at this position (see *Figure 4E* red arrow), and its 8 nm periodicity is in the range of estimated step size for dynein and kinesin motors (*Kikkawa, 2013*; *Reck-Peterson et al., 2006*; *Coy et al., 1999*).

Two holes were observed in the inner junction between the A- and B-tubules (red arrows in *Figure 3C*). We termed these 'proximal' and 'distal' holes, based on their position relative to the proximal end of the axoneme. The distal hole is near the site of NDRC attachment to the DMT and corresponds to the hole reported in other organisms (*Nicastro et al., 2011*; *Pigino et al., 2012*). The distal hole in *Chlamydomonas* is dependent on the presence of the NDRC on the external face of the DMT (*Heuser et al., 2012b*). The proximal hole is specific to *T. brucei*. Unlike the distal hole, there are no structures on the external face of the DMT at the site where the proximal hole is located. This

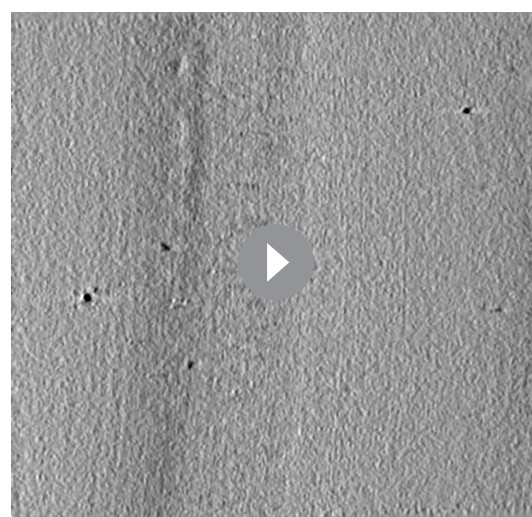

**Video 3.** Slices through a representative tomogram reconstructed by simultaneous iterative reconstruction technique (SIRT) of BSF *T. brucei*.
https://elifesciences.org/articles/52058#video3

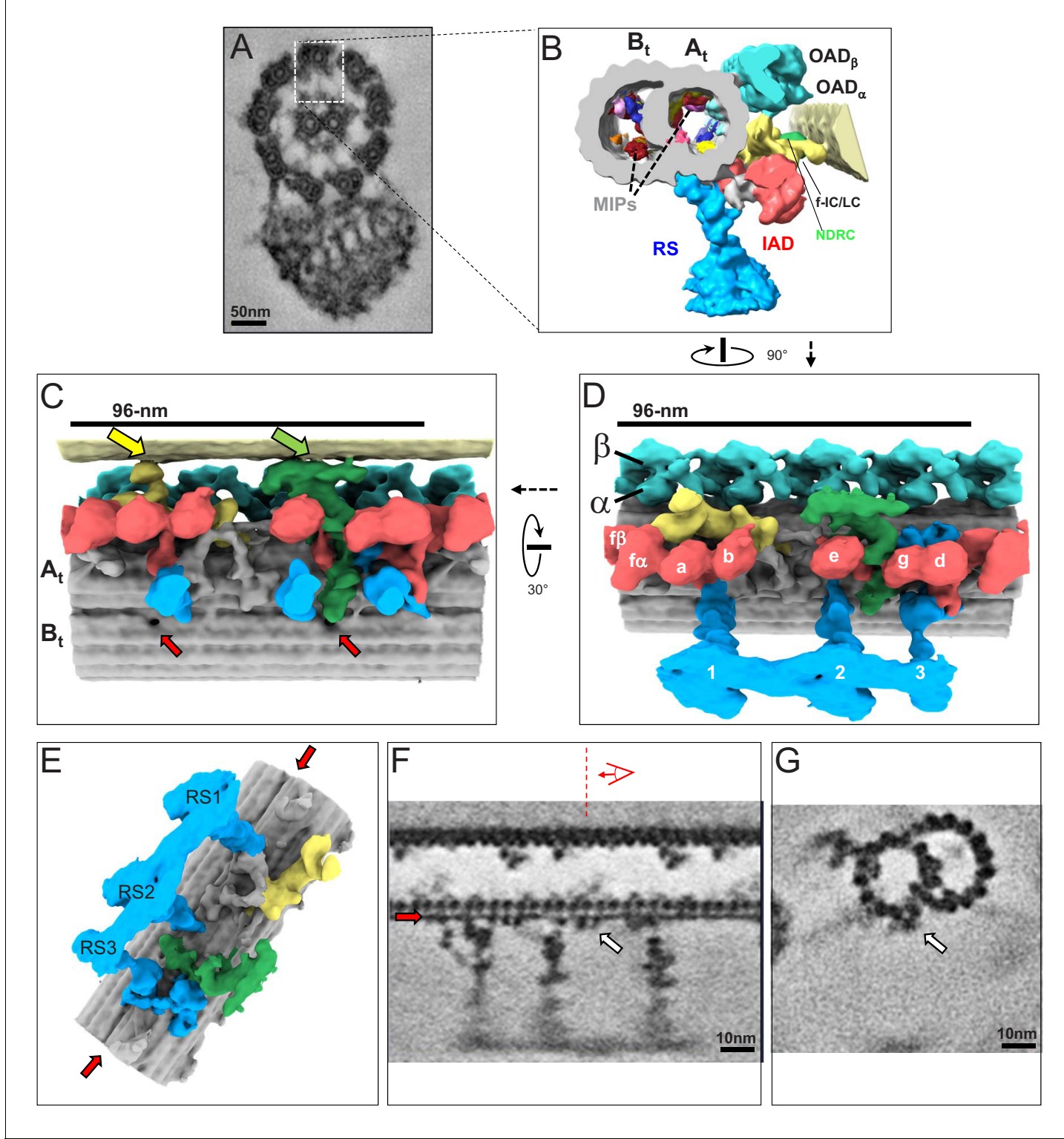

**Figure 3.** The 3D ultrastructure of the 96-nm repeat from intact axonemes of BSF *T. brucei*. (A) A representative cross-section of a demembranated and negative-stained *T. brucei* flagellum, viewed from the proximal end (adapted from *Hughes et al., 2012*). Boxed region orients the view of the averaged 96-nm repeat along a DMT shown in B. (B) Cross-section view of the 96-nm repeat obtained by sub-tomogram averaging. Labeled are: the A- and B-tubule (At, Bt), Microtubule Inner Proteins (MIPs), Radial Spokes (RS), Inner Arm Dyneins (IAD), Nexin Dynein Regulatory Complex (NDRC), IAD-f-Intermediate Chain/Light Chain Complex (f-IC/LC), Outer Arm Dynein (OAD). The surface of the B-tubule from the adjacent DMT is visible on the right. The coloring scheme is as follows: cyan, OAD; red, IAD; blue, RS; green, NDRC; yellow, dynein f IC/LC. This scheme is consistently used throughout all

*Figure 3 continued on next page*

*Figure 3 continued*

main figures, figure supplements and videos unless stated otherwise. (C, D) Shaded surface rendering longitudinal views of the 96-nm repeat. Panel C shows the view from the center of the axoneme looking outward with the proximal end of the axoneme on the left and spoke heads removed for clarity (rotation relative to Panel D is shown). The surface of the B-tubule of the adjacent DMT is visible on top. Yellow and green arrows point to the inter-doublet connections formed by the f-connector and NDRC, respectively. Red arrows point to the proximal and distal holes in the inner junction between the A- and B-tubules. Panel D shows the view from the adjacent DMT, with proximal end of the axoneme on the left (rotation relative to panel B is shown). For reference, alpha (α) and beta (β) OAD are indicated, individual IADs and RS are labeled. (E) Shaded surface rendering of the averaged 96-nm repeat with the IAD, OAD and MIA complex removed, showing a massive structure at the base of the RS3 (see also *Figure 3—figure supplement 1*). Red arrows point to the density corresponding to the FAP59/172, 96-nm ruler (*Oda et al., 2014a*) between protofilaments A2 and A3. (F, G) Longitudinal (F) and transverse (G) density slices of the averaged 96-nm repeat. Red arrows in panels E and F point at the density of the FAP59/172 ruler between protofilaments A2 and A3. The red dashed line and perspective cartoon in panel F show the position and perspective of the cross-section shown in G, with the white arrow in panels F and G indicating the FAP59/172 ruler.

The online version of this article includes the following source data and figure supplement(s) for figure 3:

**Figure supplement 1.** Cross sections of the ten tomograms used to obtain the entire averaged BSF 96-nm axonemal repeat structure.
**Figure supplement 2.** Gold Standard Fourier shell correlation (FSC) and ResMAP analyses.
**Figure supplement 3.** Extra densities outside protofilaments b7b8 and massive density at the base of RS3 in BSF *T. brucei*.
**Figure supplement 4.** Illustration of the principles of *autoPicker*.
**Figure supplement 4—source code 1.** Source code for subtomogram averaging method.

indicates the proximal hole reflects structural properties imparted by proteins of the inner junction or inside the microtubules and is not dependent on the presence of external structures.

Interconnections were observed between substructures on the A-tubule, including between individual OADs (*Figures 3D* and *4*), between OAD and the IAD-f complex (*Figure 3B,D*, *Figure 3—figure supplement 3A,D*). Particularly noteworthy are extensive contacts between RS3, IAD-d, and the A and B-tubules (*Figure 3C*, *Figure 3—figure supplement 3C,D*). At the base of RS3 we observed a structure that extends over four A-tubule protofilaments and attaches to the inner junction. Unlike the case for *Chlamydomonas* (*Nicastro et al., 2006*), the NDRC did not make direct contact with the OAD in *T. brucei* (*Figure 3—figure supplement 3D*), suggesting differences in mechanisms for coordinating inner and outer dynein motor activities.

## Axonemal dynein arrangement in *T. brucei*

An earlier cryoET study of the *T. brucei* axoneme revealed the expected 4 OADs/repeat but did not resolve individual dynein motors (*Hughes et al., 2012*). With sub-tomogram averaging, the beta and alpha OAD motors are now clearly resolved (*Figures 3B,D* and *4A*). This result provides the first direct demonstration that OADs contain two motor domains in *T. brucei*, making it the first protist shown to have two motors per OAD and correcting a misconception that all protists contain three motors (*Lin and Nicastro, 2018*). Together with three radial spokes per repeat, the entire arrangement of the *T. brucei* axoneme determined here therefore resembles that of humans more so than does *Chlamydomonas* or *Tetrahymena*, which are used as models for human cilium structure and function (*Figure 5*) (*Pigino et al., 2012*; *Owa et al., 2019*; *Lin et al., 2014*).

Axoneme motility is driven by rotation of the dynein AAA+ ring relative to the linker and tail domains, causing translocation of adjacent DMTs as the dynein transitions from pre-powerstroke to post-powerstroke position (*Lin and Nicastro, 2018*; *Kikkawa, 2013*; *Burgess et al., 2003*). The AAA+ ring, linker and tail domains were resolved in the OAD-beta dynein and are in the post-power stroke position (*Figure 4B,C*), consistent with the fact that samples were prepared without exogenous ATP. This result thus supports structural assignments in the averaged structure. The dynein stalk domain, which contacts the adjacent DMT is visible (*Figure 4E*).

Six IADs were well-resolved (*Figure 3C,D*) and annotated f, a, b, e, g, and d, according to standard nomenclature (*Bui et al., 2012*). Notably, IAD-c, which is important for movement of *Chlamydomonas* in high viscosity (*Yagi et al., 2005*), is absent from the trypanosome structure. This finding is notable, given the very viscous environments experienced by trypanosomes during movement through tissues of the mammalian host (*Heddergott et al., 2012*; *Bargul et al., 2016*; *Capewell et al., 2016*; *Trindade et al., 2016*) and tsetse fly vector (*Schuster et al., 2017*).

## Extensive Inter-doublet connections in the *T. brucei* axoneme

Nexin links are connections between adjacent DMTs, that are visible in axoneme TEM thin sections. They stabilize the axoneme and are a fundamental component of the sliding filament model for axoneme motility (*Satir, 1968*; *Satir et al., 2014*; *Viswanadha et al., 2017*). Prior studies indicate the NDRC is the only nexin link in *Chlamydomonas* (*Figure 5A*) (*Heuser et al., 2009*). In *T. brucei*, however, we identified two prominent inter-doublet connections, the NDRC and the IC/LC complex of IAD-f (*Figure 3B–D*). We term this second connection the 'f-connector'. The NDRC and f-connector each extend from the A-tubule of one DMT to contact near protofilament B9 of the adjacent DMT. NDRC contact is through the proximal and distal lobes defined by *Heuser et al. (2009)* and extends approximately 31 nm. The f-connector contact region extends approximately 11 nm. A structure analogous to the f-connector is observed between neighboring DMTs of three specific DMT pairs in *Chlamydomonas* (*Bui et al., 2009*). However, the prominence of the f-connector observed here in *T. brucei* suggests it is present between neighboring DMTs of most and perhaps all DMTs, a conclusion supported by analysis of individual DMTs (see below), indicating that nexin links in *T. brucei* include both the NDRC and the f-connector, as well as the OAD inter-doublet connector described below. This distinguishes the *T. brucei* axoneme from 3D axoneme structures from other organisms so far reported (*Figure 5*) (*Pigino et al., 2012*; *Owa et al., 2019*; *Lin et al., 2014*).

A conspicuous structure not previously reported in any organism is a large protrusion at the junction between the tail and stalk domains of OAD-alpha (*Figure 4D,F*). This protrusion, which we termed the 'OAD inter-doublet connector', extends to the space between protofilament B6 and B7 of the adjacent DMT. The OAD inter-doublet connector is thus distinguished from the OAD-alpha stalk, which extends from the AAA+ ring to the space between protofilament B7 and B8 of the adjacent DMT (*Figure 4E*). The OAD inter-doublet connector is present on all four OAD-alpha motors in the 96-nm repeat but is not observed in OAD-beta.

## Doublet-specific features of the 96-nm repeat

The 96-nm repeat structure described above represents an average of all nine DMTs and does not reflect heterogeneity that may distinguish individual DMTs, as reported for *Chlamydomonas* (*Bui et al., 2012*). To address this, we did sub-tomogram averaging on each DMT separately. The PFR restricts axoneme orientations on the EM grid and consequently, individual DMT structures suffer from the missing wedge. This was most severe for DMT 3 and 7 and we therefore cannot comment on these DMTs (*Figure 6—figure supplement 1F–G*). For the remaining seven DMTs, distortion due to the missing-wedge problem obscured some details, particularly MIPs and OADs. However, main features of the 96-nm repeat were resolved (*Figure 6—figure supplement 1B–E*). Each DMT was distinct, but careful examination revealed some similarities, particularly in the region of IAD-b, between DMTs 1+5, 2+6 and 8+9 (*Figure 6—figure supplement 1C–E*). Therefore, to reduce the impact of the missing wedge, we averaged DMTs within these pairs together. We recognize that this approach may still mask some features of a single doublet, but it nonetheless reveals heterogeneity between doublets.

As shown in *Figure 6* and *Figure 6—figure supplement 1*, we identified doublet-specific structures that were not evident in the entire averaged structure. DMT 8 and 9 are distinguished from all other DMTs in that they do not have an IAD-b. In the place of IAD-b is a previously undescribed

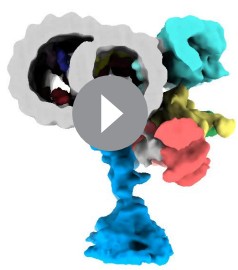

**Video 4.** 3D surface rendering of the averaged 96-nm axonemal repeat from BSF *T. brucei*, rotated to show the structures of DMT (grey), Radial spokes (blue), NDRC (green), f IC/LC (yellow) and OAD (cyan) and IAD (red).
https://elifesciences.org/articles/52058#video4

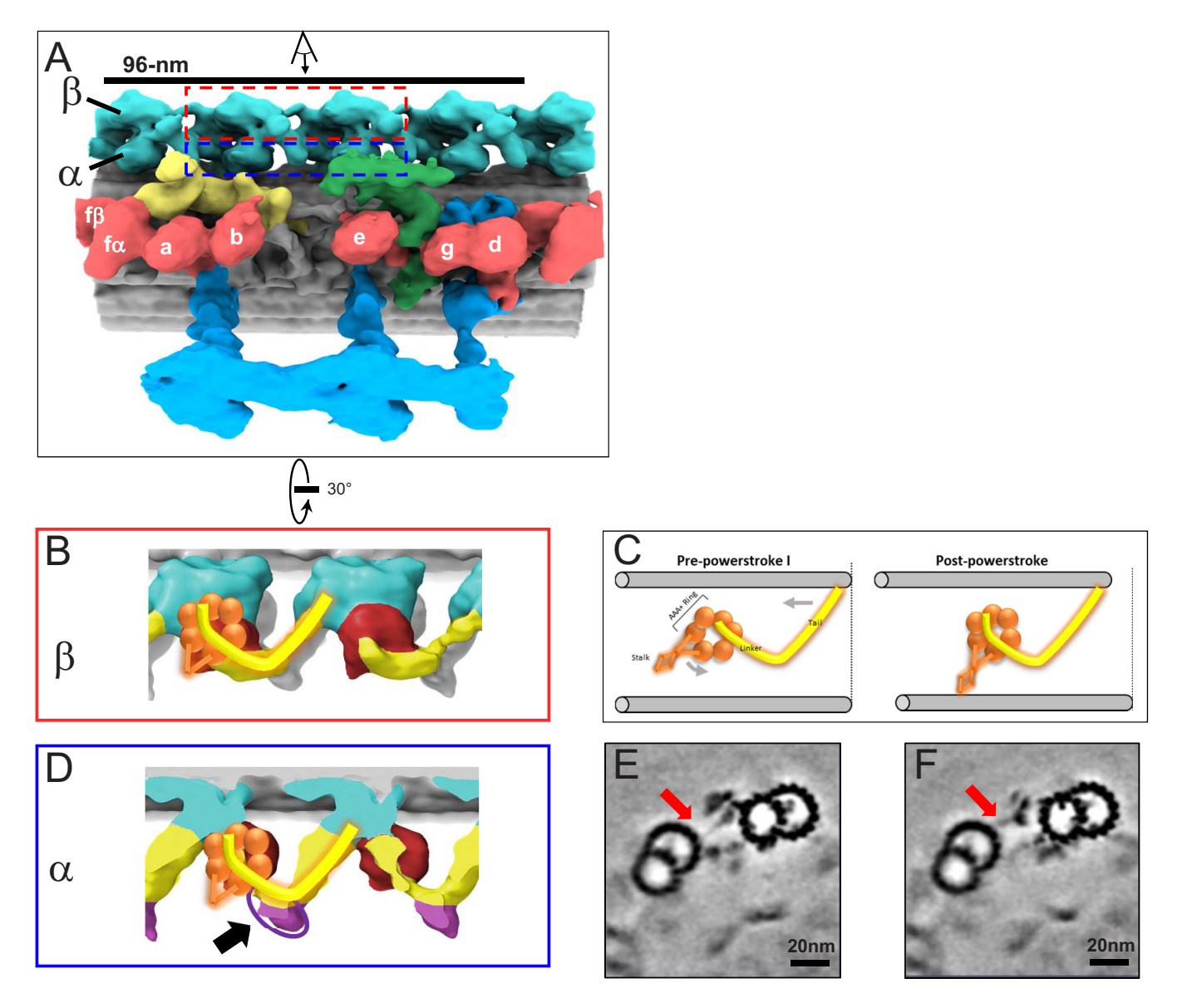

**Figure 4.** In situ structure of outer arm dyneins and novel OAD-alpha inter-doublet connector in BSF *T. brucei*. (A) Shaded surface rendering, longitudinal view of the averaged 96-nm repeat. Coloring as described for *Figure 3A*. The box around the OAD indicates the region and perspective shown in B (red box) and D (blue box). (B, D) Shaded surface renderings of outer arm dyneins from the averaged 96-nm repeat. (B) Two adjacent OADβ dyneins. The linker and tail domains are colored yellow and the AAA+ ring is red. Cartoon overlay shows the post-powerstroke position of dynein. (D) Top view of two adjacent OADα dyneins. The linker and tail domains are colored in yellow and the AAA+ ring is colored in red. The arrow points to the OADα connector (purple), at the junction between the tail and linker domains. Cartoon overlay shows the post-powerstroke position of dynein. (C) A schematic illustrating relative DMT movement as dynein moves from pre-powerstroke one state (left) to post-powerstroke state (right). (E–F) Density slices of the averaged 96-nm repeat, viewed in cross-section, viewed from the distal tip of the axoneme. Red arrows indicate the dynein stalk domain in (E), and the OADα connector in (F), contacting the neighboring DMT.

arch-like structure that extends upward from between RS1 and RS2, which we termed 'arch' (*Figure 6D*). DMT 1 and 5 are distinguished by the presence of a novel inter-doublet connecter, which we termed 'b-connector', that connects IAD-b to the adjacent DMT and includes a 'tail' domain that connects with the 'Modifier of Inner Arms' MIA complex (*Yamamoto et al., 2013*) (*Figure 6B*). DMT 2 and 6 contain a b-connector that lacks the tail domain (*Figure 6C*). DMT 4, 8 and 9 lack the b-connector. Structural variation of the b-connector on different DMTs explains why it

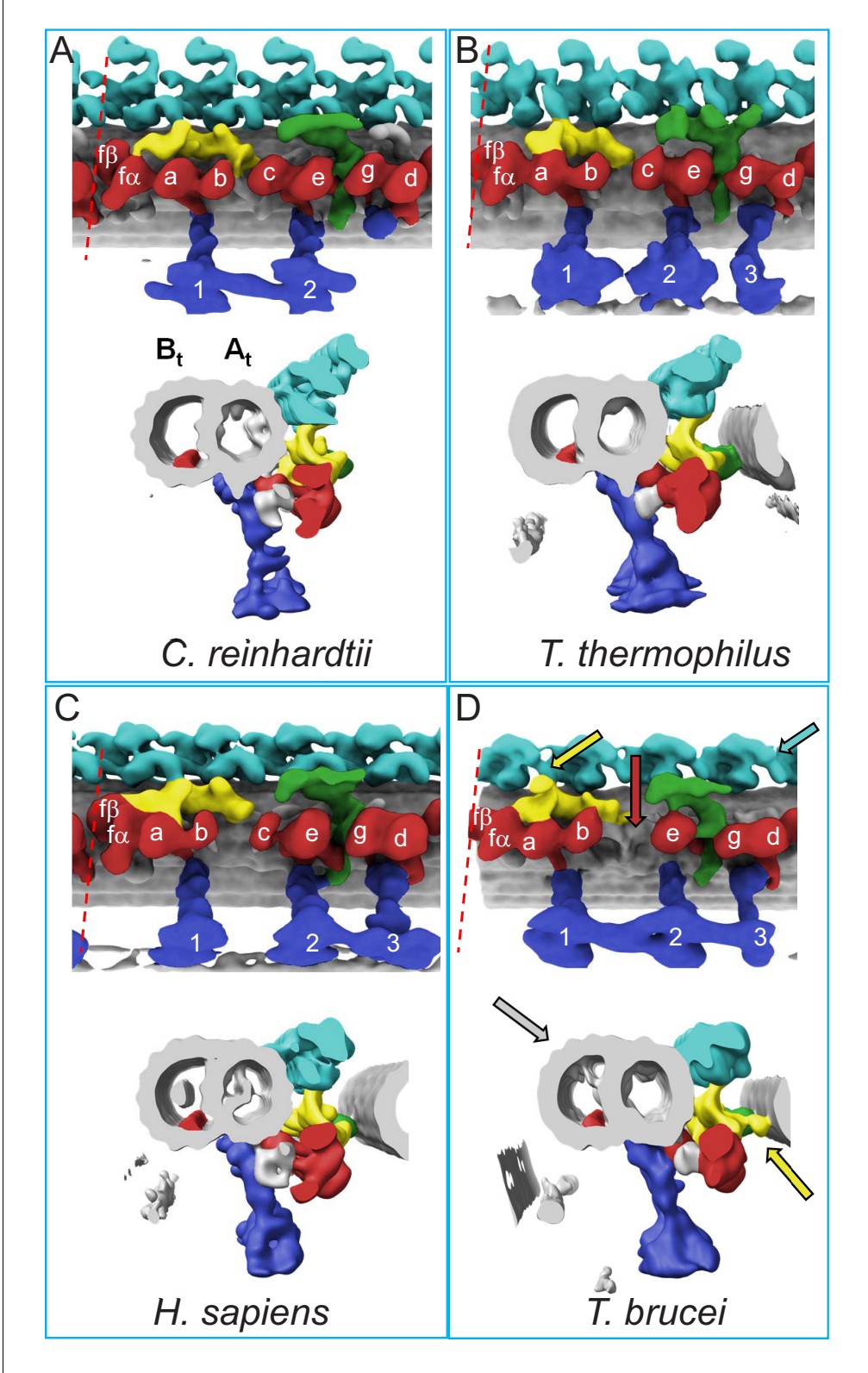

**Figure 5.** Comparison of 96-nm axonemal repeat structures across species. (**A–D**) Structure of the 96-nm axonemal repeat is shown for *Chlamydomonas reinhardtii* (**A**) (***Owa et al., 2019***), *Tetrahymena thermophilus* (**B**) (***Pigino et al., 2012***), *Homo sapiens* (**C**) (***Lin et al., 2014***) and BSF *Trypanosoma brucei* (**D**) (this work). Longitudinal (top) and cross-sectional (bottom) views are shown for each. Canonical features of the 96-nm repeat

*Figure 5 continued on next page*

*Figure 5 continued*

are colored, including outer arm dyneins (cyan), inner arm dyneins (red and numbered according to convention) the IC/LC complex of inner arm dynein f (yellow), the NDRC inter-doublet linkage (green) and radial spokes (blue). The microtubule lattice is gray and the A- and B-tubules are indicated. MIP3 (red) is present in all organisms shown and is colored in the B-tubule for reference. For all structures except that from *C. reinhardtii*, the surface of the B-tubule from the adjacent DMT is shown. Inner dyneins and radial spokes are labeled for reference. The red dashed line indicates the position of viewing for the cross-section shown. All structures are filtered to resolution of 50 Å. Features that distinguish the *T. brucei* repeat include the f-connector (yellow arrow), missing dynein-c (red arrow), lineage specific MIPs within the A- and B-tubules (gray arrow), and two OAD motors in a protist (cyan arrow). Other *T. brucei*-specific structures, such as the OAD-alpha inter-doublet connector and b-connector are not visible in this view.

was not evident in the entire averaged structure. DMTs 1, 4, 5, 6, 8 and nine each have an

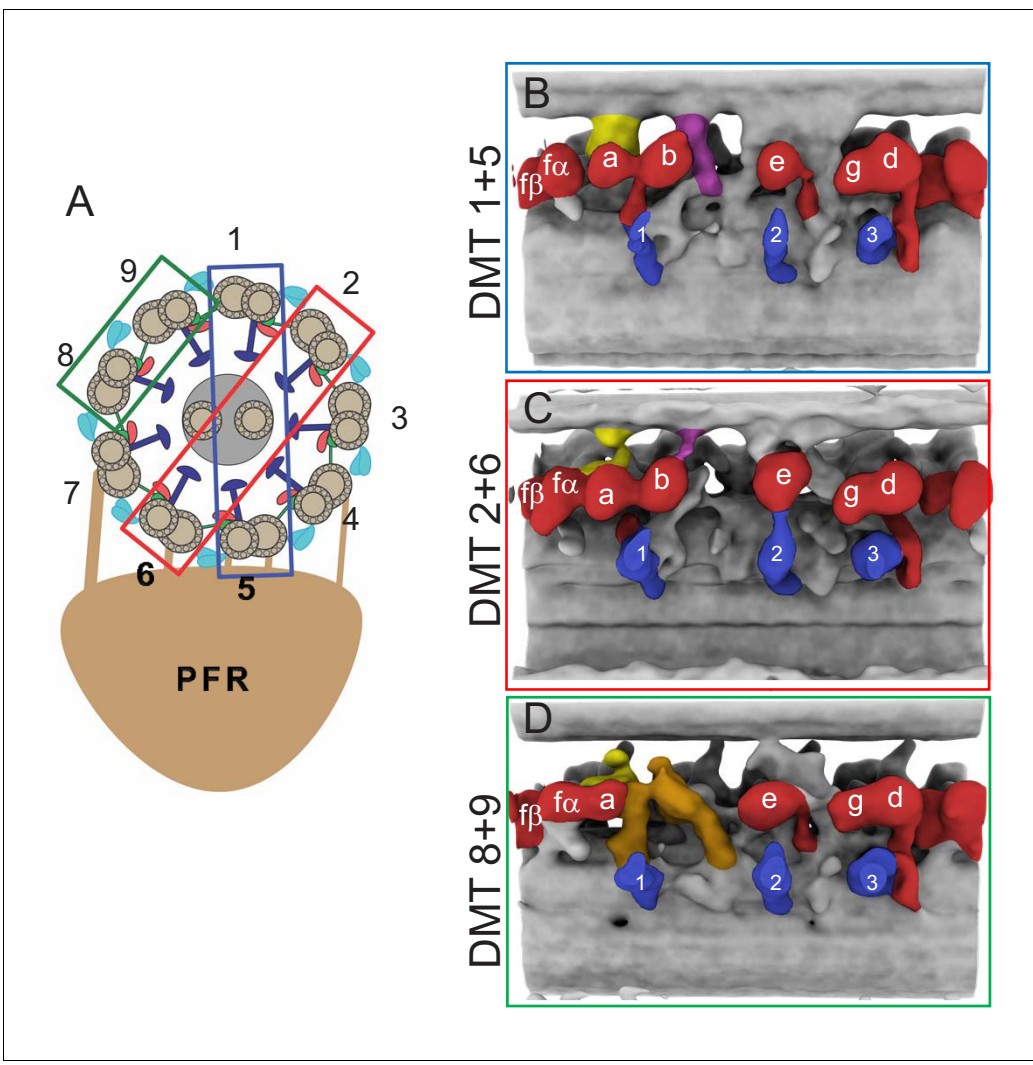

**Figure 6.** Doublet-specific structures of the BSF *T. brucei* 96-nm repeat. (**A**) Schematic showing the numbering of individual DMTs. Colored boxes indicate the DMT pairs that were used for the averaged structures shown in panels B-D. (**B–D**) Panels show averaged structures for DMT pairs 1+5 (**B**), 2+6 (**C**), and 8+9 (**D**). Inner arm dyneins (red) and radial spokes (blue) are labeled for reference. The f-connector, b-connector and the arch that distinguish DMTs 8 and 9 are colored yellow, purple and brown, respectively.

The online version of this article includes the following figure supplement(s) for figure 6:

**Figure supplement 1.** Sub-tomogram averages of the 96-nm repeat of individual DMTs of BSF *T. brucei*.

f-connector structure. DMT two does not have a clear f-connector, but this may reflect a missing wedge artifact since the density of the NDRC connection is also reduced (*Figure 6—figure supplement 1D*). The analysis of individual DMTs supports the interpretation that the f-connector is present on most DMTs. Additionally, this analysis identified a new lineage specific inter-doublet connection not present in other organisms, the b-connector.

The PFR is attached to DMT 4, 5, 6 and 7 and we therefore considered whether this attachment alters the 96-nm repeat. As detailed above, two PFR-attached DMTs, DMT 5 and 6, each show similarities to non-attached DMTs, DMT 1 and 2, that are not shared by each other (*Figure 6—figure supplement 1A,C–D*). Therefore, PFR attachment does not seem to correlate with specific structural changes in the 96-nm repeat, at least at the current resolution. PFR-attachment complexes themselves, have a 56 nm periodicity (*Hughes et al., 2012*; *Koyfman et al., 2011*) and therefore would not be resolved in our 96-nm repeat structure.

## CMF22/DRC11 is part of the NDRC proximal lobe involved in binding the adjacent DMT

The NDRC functions in axoneme stability and motility and these functions are thought to be mediated in part through inter-doublet connections (*Viswanadha et al., 2017*; *Olbrich et al., 2015*; *Wirschell et al., 2013*; *Ralston and Hill, 2006*). The NDRC is composed of at least 11 subunits and some of these have been positioned within the complex (*Heuser et al., 2009*; *Yamamoto et al., 2013*; *Ralston et al., 2006*; *Nguyen et al., 2013*; *Kabututu et al., 2010*; *Bower et al., 2013*; *Lin et al., 2011*; *Huang et al., 1982*; *Song et al., 2015*; *Oda et al., 2014b*). However, subunits that contact the B-tubule of the adjacent DMT are unknown. We identified CMF22 as a subunit of the *T. brucei* NDRC (*Nguyen et al., 2013*), and the *Chlamydomonas* CMF22 orthologue is DRC11 (*Bower et al., 2013*). RNAi knockdown of CMF22/DRC11 abolishes forward motility in *T. brucei*, demonstrating the importance of DRC11 in axoneme motility (*Video 5* and *Video 6*) (*Nguyen et al., 2013*). The position of CMF22/DRC11 in the NDRC is unknown, but biochemical data indicate it may be within the proximal or distal lobe structures that contact the adjacent DMT (*Nguyen et al., 2013*; *Bower et al., 2013*; *Awata et al., 2015*). We therefore used cryoET and sub-tomogram averaging to determine the structural basis of the CMF22/DRC11 RNAi knockdown. We used procyclic culture form (PCF) *T. brucei*, because loss of axonemal components is lethal in bloodstream forms (*Ralston and Hill, 2006*; *Broadhead et al., 2006*; *Ralston and Hill, 2008*).

The 96-nm repeat of WT PCF (*Figure 7A*) axonemes was very similar to that of BSF (*Figures 3* and *4*), including the presence of the novel OAD inter-doublet connector and the f-connector, as well as the missing IAD-c. In the CMF22 knockdown, the only structure clearly affected is the NDRC (*Figure 7C–E*). The entire structure of the complex is mostly preserved, but the proximal lobe of the linker region is severely reduced (*Figure 7E*). The affected structures encompass a large portion of the inter-doublet contact area for the *T. brucei* NDRC and include both regions reported to contact the adjacent DMT in the *Chlamydomonas* NDRC (*Heuser et al., 2009*). The remaining NDRC domains, including dynein contacts were not grossly affected, although connection from NDRC to the MIA complex (*Yamamoto et al., 2013*) might be altered. Therefore, inter-doublet connection mediated by the NDRC is critical for axoneme motility.

## Extensive, lineage-specific MIPs in *T. brucei*

One major advance resulting from cryoET studies is the discovery that protein structures inside the microtubule, first observed in trypanosomes based on transmission EM studies more than fifty years ago (*Vickerman, 1969*; *Anderson and Ellis, 1965*), are ubiquitous in axonemal microtubules (*Nicastro et al., 2011*; *Ichikawa et al., 2017*). A striking feature of *T. brucei* axonemal

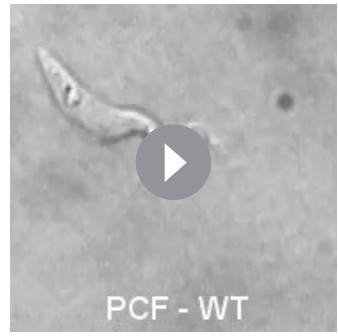

**Video 5.** Real-time video showing wild type motility of a PCF *T. brucei* parasite in culture medium. The parasite translocates using a helical movement with flagellum tip leading.
https://elifesciences.org/articles/52058#video5

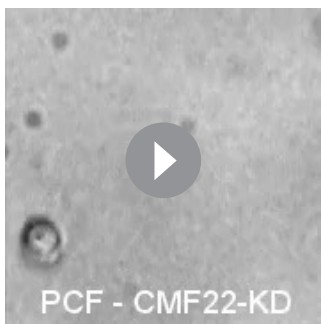

**Video 6.** Real-time video of a CMF22-knockdown PCF *T. brucei* parasite in culture medium. The flagellum beats but is unable to drive translocation of the parasite.
https://elifesciences.org/articles/52058#video6

microtubules is the presence of extensive MIP complexes not only in the A-tubule, but also in the B-tubule (*Figures 3B* and *8A* and *Supplementary file 1*). *Figure 8A* shows a cross-section view of the averaged 96-nm repeat looking from the proximal end of the axoneme, with MIPs colored and external structures removed for clarity. The B-tubule is on top and the A-tubule is below, with 13 protofilaments of the A-tubule and 10 protofilaments of the B-tubule labeled according to convention (*Figure 8A*). The shape, position and periodicity of the structure inside the B-tubule, next to the inner junction between the A- and B-tubules (*Figure 8A,B*), indicate that this structure corresponds to MIP3 described in other organisms (*Nicastro et al., 2011*; *Ichikawa et al., 2017*). Notably however, the relationship of other MIPs in *T. brucei* to previously described MIPs is unclear and most TbMIPs in both the A- and B-tubules appear to be trypanosome-specific (*Figure 5*).

When viewed in longitudinal section from within the B-tubule, TbMIP3 consists of two lobes, 3a and 3b (*Figure 8B*), as reported for *Chlamydomonas* and *Tetrahymena* (*Nicastro et al., 2011*; *Ichikawa et al., 2017*). There are six such TbMIP3 structures in each 96-nm repeat. Subtle structural variations in the sizes of lobe 3b and connections to lobe 3a yield a 48 nm repeating pattern of three adjacent TbMIP3 structures, colored red, gold and orange (*Figure 8B*). These TbMIP3 variations coincide with other structural variations within the microtubule, such as presence of inner junction holes (arrows in *Figure 8B*), unique contacts to Snake MIP (see Snake MIP description below), and attachment to a structure identified as MIP3c in *Chlamydomonas* (*Owa et al., 2019*) (asterisks in *Figure 8B*). Variation in lobe 3b between the two gold TbMIP3 structures could suggest a 96-nm repeat unit, but this variation probably results from interference from the DRC base plate on the outside of the DMT at the site of the distal hole.

Facing TbMIP3, on the opposite side of the B-tubule lumen, are several trypanosome-specific MIPs, MIP B5, B4, B2 and a MIP that extends across the entire lumen, thus corresponding to the ponticulus structure previously observed in classical thin section TEM (*Figure 8C*) (*Vickerman, 1969*; *Anderson and Ellis, 1965*; *Vaughan et al., 2006*). To our knowledge, the ponticulus was the first structure observed within the microtubule lumen in any organism and is the only structure so far described to extend across the entire microtubule. Our 3D structure shows that the ponticulus is not a single structure, but rather is comprised of 3 discrete MIPs, which we termed Pa, Pb and Pc (*Figure 8C–F*). Each ponticulus MIP extends across the entire B-tubule lumen, connecting the A-tubule lattice to a different B-tubule protofilament. Pa, Pb and Pc connect protofilament A12 to protofilaments B3, 5 and 4, respectively and exhibit 48 nm periodicity (*Figure 8C–F*). The ponticulus is assembled after construction of the axoneme (*Vaughan et al., 2006*). Therefore, proteins comprising these structures must be delivered into a fully formed DMT.

The A tubule also contains a diverse cohort of MIPs each with a repeating unit of 48 nm (*Supplementary file 1*, *Figure 8A*, *Figure 9—figure supplement 1*). Rather than constituting several isolated structures however, TbMIPs form a network of interconnected complexes, similar to, but more extensive than, that reported for *Tetrahymena* (*Ichikawa et al., 2017*). Two A-tubule MIPs are particularly notable. One, which we termed 'ring MIP', is unique among MIPs so far described because it forms a ring structure protruding into the microtubule lumen (*Figure 9B*). The ring MIP is attached to the protofilaments A8 and 9 and contacts another MIP complex on the protofilaments A8-12 termed 'Ring Associated MIP' (RAM) (*Figure 9B,C*). Another MIP, which we termed 'snake MIP', presents as a serpentine structure that appears to weave in and out of the A and B-tubules (*Figure 10* and *Video 7*). The continuity of this density suggests it might be a contiguous structure, extending 48 nm and spanning multiple tubulin subunits, although we cannot rule out the possibility that protofilament subunits contribute to this structure.

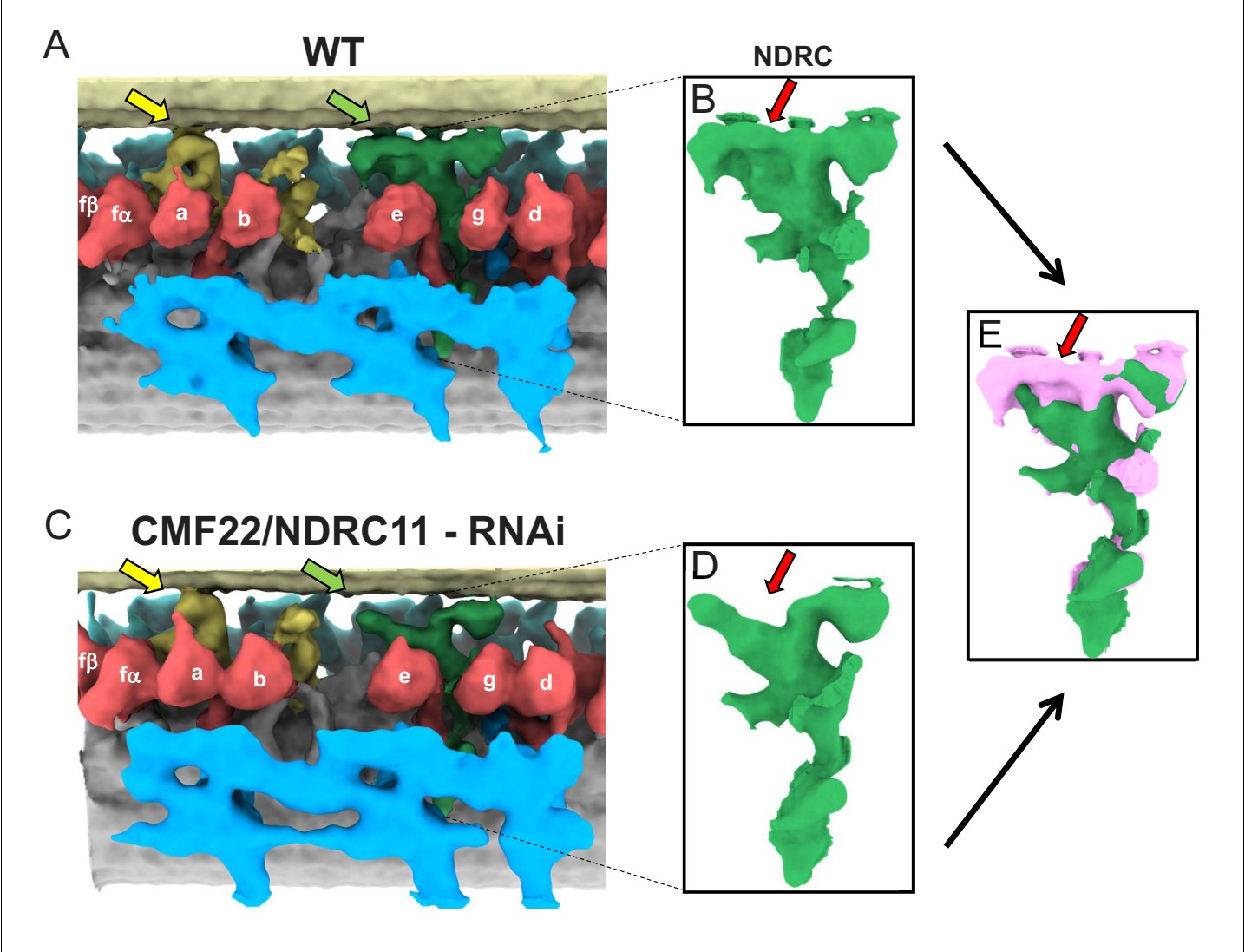

**Figure 7.** Comparison between averaged 96-nm repeats of wild-type and CMF22/DRC11 knockdown PCF *T. brucei*. (**A, C**) Sub-tomogram averages of the 96-nm repeats of wild-type (**A**), and CMF22/DRC11 knockdown mutant (**C**). Yellow and green arrows point to the region of the B-tubule contacted by the f-connector and NDRC, respectively. (**B, D**) Zoomed-in view of the NDRC from WT (**B**) and CMF22/DRC11 knockdown (**D**). The red arrow in each panel denotes the structure most substantially affected in the knockdown. (**E**) Superposition of the NDRC structures shown in B and D, with WT in pink and the mutant in green. The red arrow indicates the most striking difference, corresponding to inter-doublet contacts made by the NDRC.

## Discussion

The ciliary axoneme is one of the most iconic features of eukaryotic cells and is considered to have been present in the last eukaryotic common ancestor (LECA) (*Khan and Scholey, 2018*). To date, however, high-resolution structures of the 96-nm axoneme repeat have only been reported for two of the three eukaryotic supergroups. Here we report the 3D ultrastructure of the *T. brucei* 96-nm axonemal repeat. This is the first such structure reported for any pathogenic organism and first representative from the eukaryotic lineage of Excavates, a basal group that includes many pathogens of global importance to human health and agriculture (*Hampl et al., 2009*; *Dawson and Paredez, 2013*). Our studies indicate the diversity of structures comprising the 96-nm repeat is under appreciated, give insight into principles of axoneme structure and function, and identify pathogen-specific features that may support unique motility needs of trypanosomes.

The genus *Trypanosoma* was discovered more than 175 years ago and named for its unique cell motility (*Gruby, 1843*), which is driven by a single flagellum. The functional unit of the eukaryotic

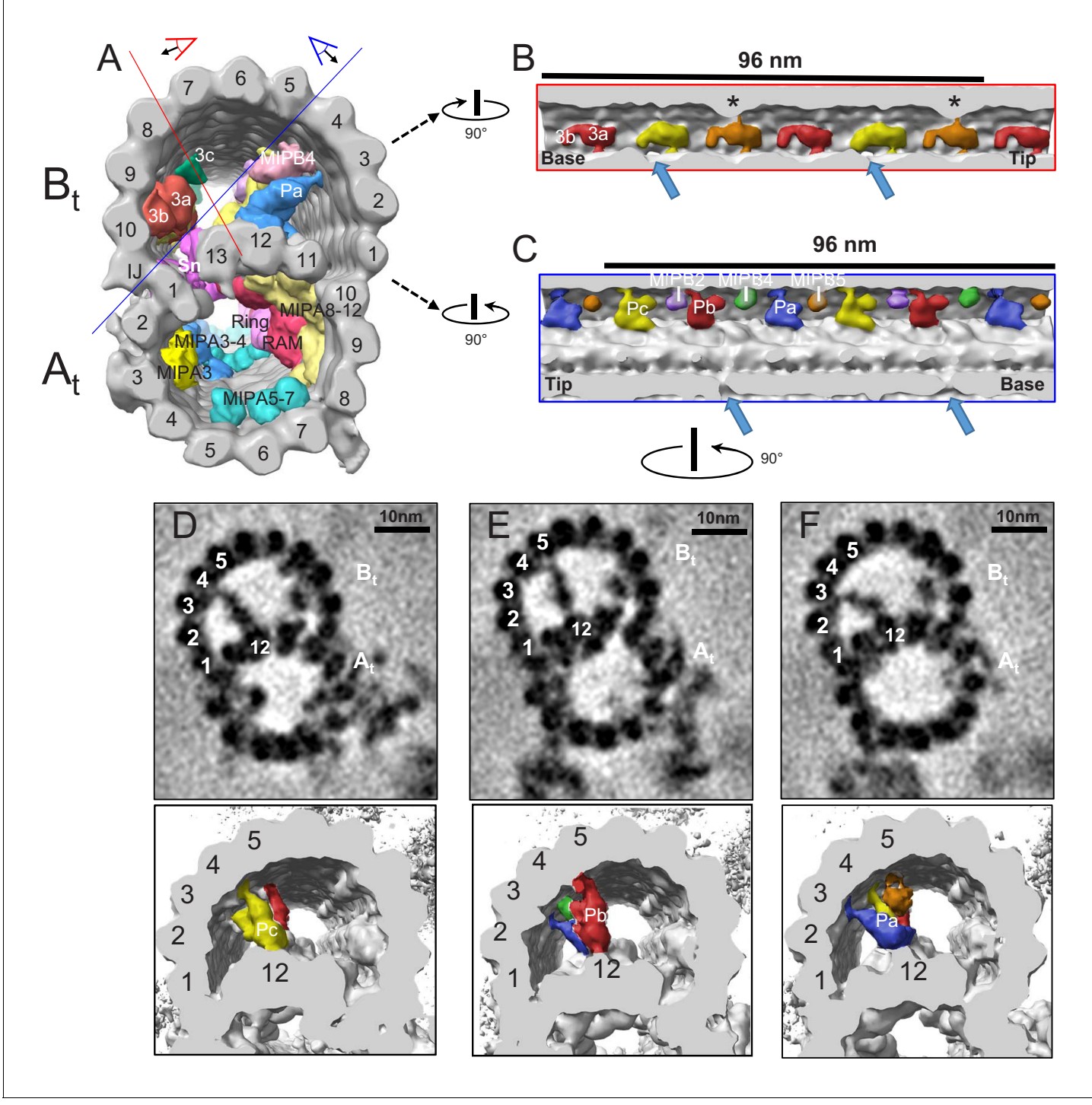

**Figure 8.** TbMIP3 and ponticulus in the B-tubule of BSF *T. brucei*. (**A**) Guide figure showing cross-section view of the averaged 96-nm repeat, viewed from the proximal end of the axoneme with MIPs colored and densities external to the DMT removed. Red and blue lines indicate sections and viewing perspectives shown in panels (**B**) and (**C**), respectively. (**B**) Longitudinal view into the inside of the B-tubule showing structural variations of TbMIP3 (red, yellow and orange) described in the text, with a periodicity of 48 nm. Arrows indicate the proximal and distal holes in the inner junction. Asterisk indicates MIP3a attachment to a structure identified as MIP3c in *Chlamydomonas* (*Owa et al., 2019*). Proximal (base) and distal (tip) ends of the repeat are indicated and rotation relative to panel A is shown. (**C**) Longitudinal view into the inside of the B-tubule showing ponticulus complexes Pa, Pb and Pc with a periodicity of 48 nm. Arrows indicate the distal and proximal holes in the inner junction and rotation relative to panel A is shown. (**D–F**) Top panels show cross-sections of average density maps viewed from the axoneme's distal tip to proximal end into the DMT. A subset of protofilaments are labeled for reference and rotation relative to panel C is shown. The trypanosome-specific Ponticulus (Pa, Pb and Pc) is seen bridging the entire lumen

*Figure 8 continued on next page*

*Figure 8 continued*

of the B-tubule from protofilament A12 to protofilaments B3, B5, and B4, respectively. The corresponding 3D isosurface renderings, looking from the same position are shown below, with Ponticulus-Pa, Pb and Pc, colored in blue, red and yellow respectively.

The online version of this article includes the following figure supplement(s) for figure 8:

**Figure supplement 1.** The Spine MIP is a contiguous structure, spanning 48 nm and contacting adjacent MIPs in BSF *T. brucei*.

flagellum is the 96-nm axonemal repeat, which encompasses dynein motors and regulatory proteins that direct flagellum beating (*Porter and Sale, 2000*). In trypanosomes, the PFR exerts influence on the axoneme (*Koyfman et al., 2011*; *Santrich et al., 1997*; *Bastin et al., 1998*), but motility is powered by the axoneme, which is the focus of the current work. Despite intense study for several decades, axoneme structures that underpin the parasite's unique mechanism of cell propulsion remained hitherto unclear. A main finding from our studies is the discovery of lineage-specific features of the *T. brucei* 96-nm axonemal repeat, including extensive and novel MIP structures and novel inter-doublet connections between adjacent DMTs (*Figures 3–6* and *8–11*). *Figure 11* shows a schematic overview of the overall 96-nm structure, previously undescribed features are labeled in panel B. We hypothesize these parasite-specific structures support unique motility needs of trypanosomes and thereby contribute to the transmission and pathogenic capacity of these organisms. The *T. brucei* axoneme is distinguished by mechanical strain experienced due to lateral attachment to the PFR and cell body, vigorous helical beating, encounter with host tissues and frequent reversals of beat direction (*Shimogawa et al., 2018*; *Koyfman et al., 2011*; *Santrich et al., 1997*; *Bargul et al., 2016*). MIPs have been shown to stabilize the axoneme in other organisms (*Owa et al., 2019*; *Ichikawa and Bui, 2018*; *Stoddard et al., 2018*) and the expanded and MIP network of *T. brucei* may therefore help maintain stability of individual DMTs. Likewise, novel inter-doublet connections are expected to help maintain axoneme integrity under these conditions, analogous to the role of NDRC inter-doublet links in maintaining alignment of DMTs in *Chlamydomonas* (*Bower et al., 2013*).

The diversity and placement of *T. brucei* MIPs are suggestive of functions beyond stability. It is difficult to imagine for example, how a ring structure like the RingMIP, protruding into the microtubule lumen, would solely provide stability. MIPs in other organisms have been demonstrated to modulate axoneme beating (*Owa et al., 2019*; *Stoddard et al., 2018*). Given the presence of numerous trypanosome-specific MIPs, together with MIP differences reported between other species (*Figure 5*), we suggest that lineage-specific MIPs may provide a mechanism for fine-tuning the beating of axonemes between species that otherwise share a basic architecture. Extra connections between DMTs can also influence axoneme beating. It has been suggested that vortical beating of nodal cilia in vertebrates axoneme may involve transmission of regulatory signals from DMT to DMT, circumferentially around the axoneme (*King, 2018*). Extensive inter-doublet connections identified in our studies provide a means for direct interaction between DMTs and could thus contribute to helical beating that is a hallmark of *T. brucei* motility. Finally, given the recent demonstration that motility is critical for *T. brucei* virulence (*Shimogawa et al., 2018*), parasite-specific features of the 96-nm repeat, which is the foundational unit of motility, may present novel therapeutic targets. Future work to identify novel *T. brucei* MIP and connector proteins will allow these ideas to be tested directly.

By defining the structural basis of the motility defect in the CMF22/DRC11 knockdown, we demonstrate a specific requirement for inter-doublet connections in axoneme motility because the defect disrupts inter-doublet connections without affecting dyneins. This contrasts to NDRC mutants analyzed previously in *Chlamydomonas*, which typically exhibit structural defects in connections to dyneins or in dyneins themselves (*Heuser et al., 2009*; *Awata et al., 2015*; *Bower et al., 2018*). An exception is *sup-pf4* (*Heuser et al., 2009*), but this mutant has only subtle effects on motility and beat frequency (*Awata et al., 2015*), which contrasts to the CMF22/DRC11 knockdown in which propulsive motility is ablated (*Nguyen et al., 2013*). Our CMF22/DRC11 knockdown studies therefore provide several important insights. Firstly, they demonstrate that penetrance of RNAi makes knockdown lines suitable for differential cryoET structural analysis in *T. brucei*. Secondly, they demonstrate CMF22/DRC11 is required for NDRC proximal lobe assembly and B-tubule attachment and, together with biochemical data (*Nguyen et al., 2013*; *Bower et al., 2013*; *Awata et al., 2015*), indicate that CMF22/DRC11 is part of the proximal lobe. Thirdly, because inter-doublet contacts are specifically

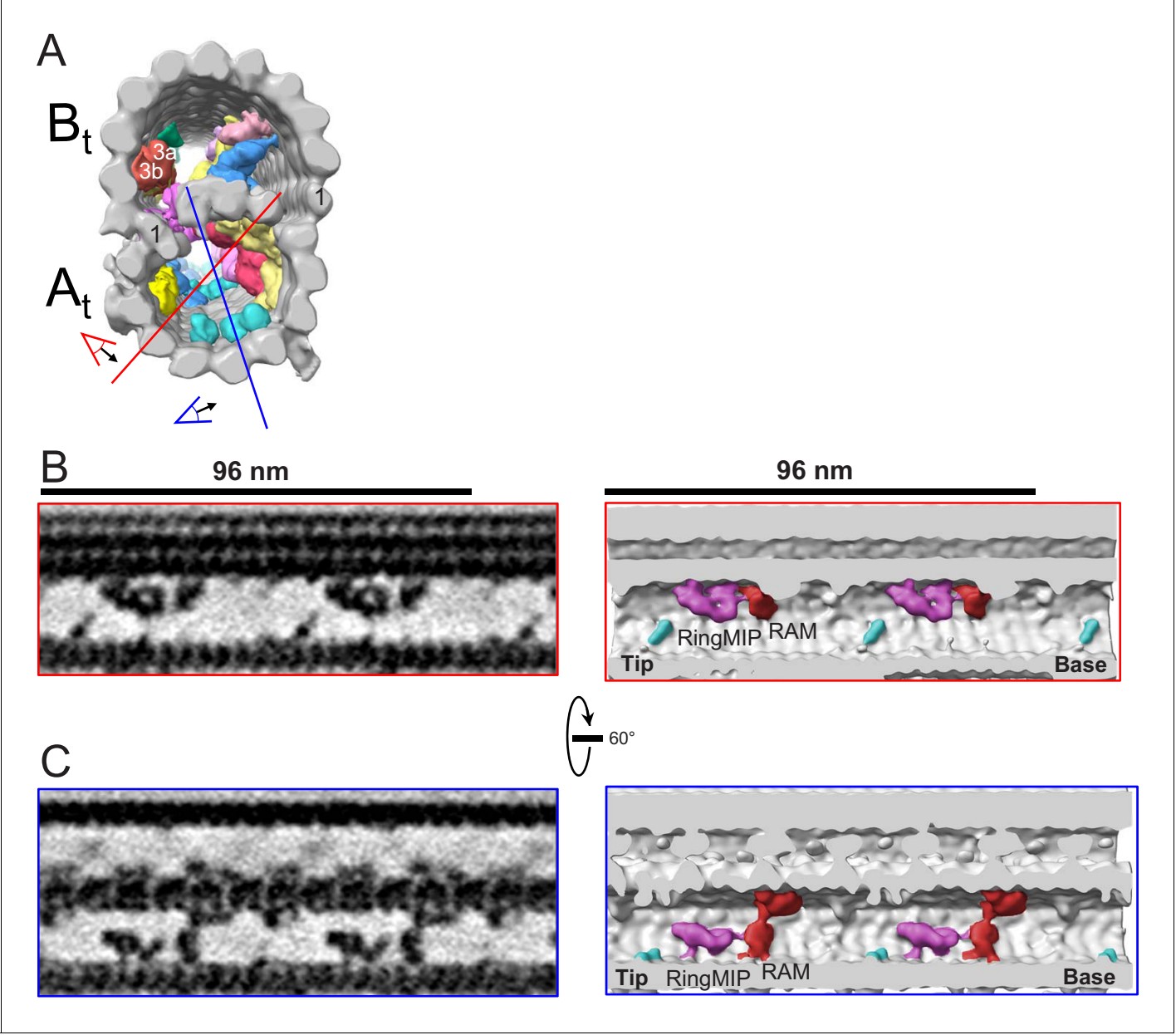

**Figure 9.** The RingMIP and Ring Associated MIP (RAM) in the A-tubule of BSF *T. brucei*. (**A**) Guide figure showing cross-section view of the averaged 96-nm repeat, viewed from the proximal end of the axoneme with MIPs colored and densities external to the DMT removed. Red and blue lines indicate sections and viewing perspectives shown in panels (**B**) and (**C**), respectively. (**B–C**) Longitudinal view of the A-tubule, showing the RingMIP and RAM. Left panels are sections through averaged density maps and right panels are corresponding isosurface renderings showing the same structures. The RingMIP (fuchsia), as well as its neighboring Ring Associated MIP (RAM) (red) and MIPA5-7 (cyan) are shown. The proximal (base) and distal (tip) ends of the axoneme are indicated and rotation of panel C relative to panel B is shown.

The online version of this article includes the following figure supplement(s) for figure 9:

**Figure supplement 1.** MIPs in the A-tubule of BSF *T. brucei*.

affected, without affecting dyneins, the results demonstrate that the NDRC itself and B-tubule contacts specifically are required for control of axoneme motility. This last point is particularly significant, as dynein-independent connection between adjacent DMTs is considered to be a founding principle of the sliding filament model for axoneme motility (*Satir, 1968*; *Holwill and Satir, 1990*; *Viswanadha et al., 2017*), yet direct tests of this idea have been limited.

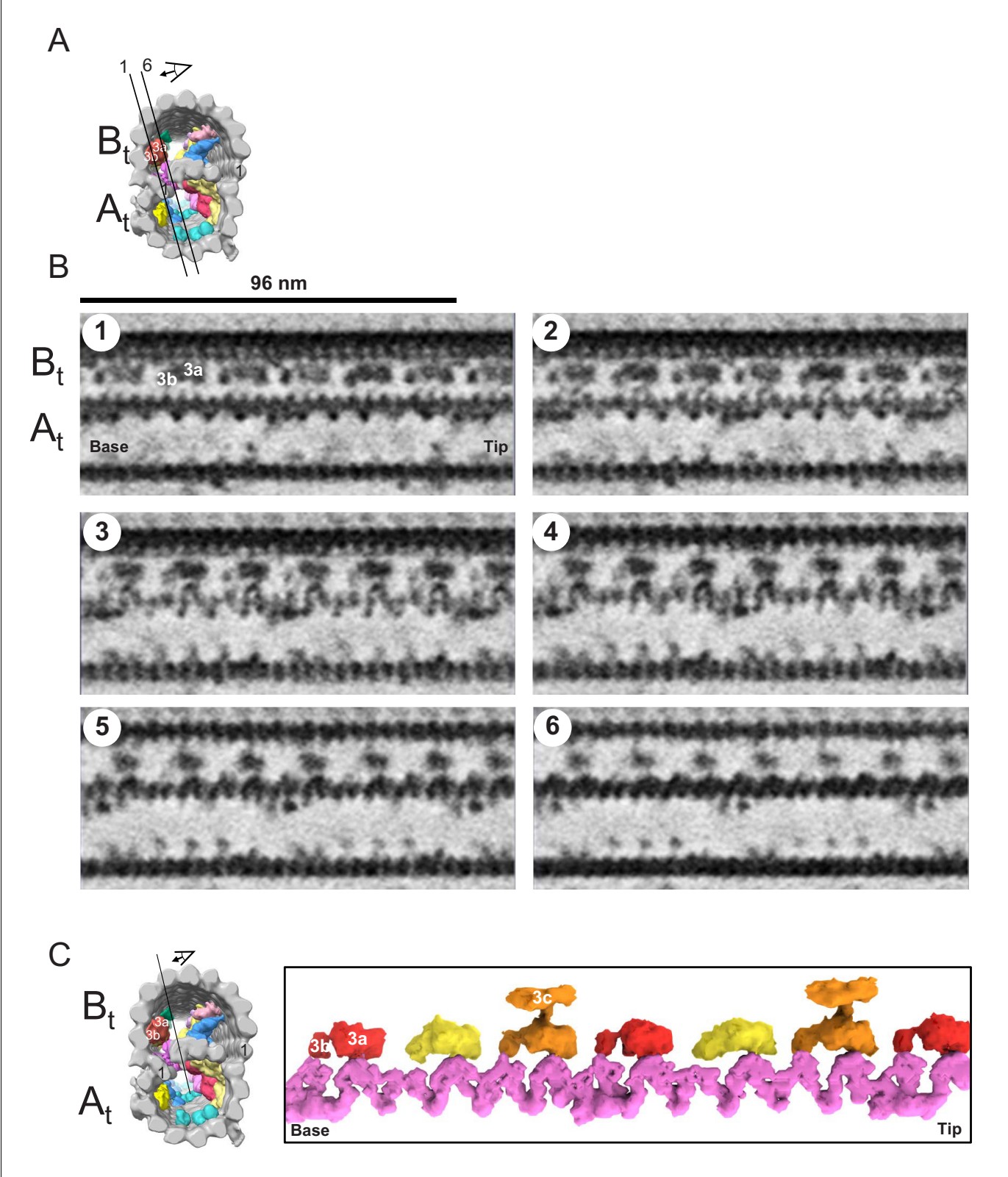

**Figure 10.** The snake MIP connects the A-tubule and the B-tubule of BSF *T. brucei*. (A) Guide figure showing cross-section view of the averaged 96-nm repeat, viewed from the proximal end of the axoneme with MIPs colored and densities external to the DMT removed. Black lines 1 and 6 show the position and perspective of sections shown in B. (B) Longitudinal view of the averaged density map. A and B-tubules are labeled. Panels 1 through 6 show six 6 Å thick, consecutive digital sections (the distance between 2 sections is 6.2 Å) through the snake MIP. (C) Left panel is a guide figure showing

*Figure 10 continued on next page*

*Figure 10 continued*

cross-section view of the averaged 96-nm repeat, viewed from the proximal end of the axoneme with MIPs colored and densities external to the DMT removed. Black line shows position and perspective for view of snake MIP shown in the right panel. Right panel shows segmented TbMIP3 (red, yellow and orange, as described for *Figure 8B*) and Snake MIP (mauve). (See also *Video 7.*).

The 96-nm spacing of the axoneme is controlled by a molecular ruler (*Oda et al., 2014a*), which is visible in the averaged BSF 96-nm repeat structure. The *T. brucei* MIP repeating unit is 48 nm, suggesting existence of a separate ruler inside the DMT to guide MIP placement. Such a ruler would need to extend 48 nm, exhibit structural heterogeneity along its length, and form contacts with other MIPs. The snake MIP satisfies these criteria. Notice, for example, that structural heterogeneities along the snake MIP coincide with unique contacts to each TbMIP3a, b structure within the 48 nm repeat (*Figure 8*). The snake MIP appears to extend into both the A- and B-tubules, which would make it possible to establish patterns in both tubules. Extensive interconnections between MIPs (*Video 7*) might allow a single ruler to guide placement of all MIPs, or there might be more than one ruler, as is suggested for the outside of DMTs in *Chlamydomonas* (*Song et al., 2018*), where the 24 nm repeat of OADs is dictated by something other than the FAP59/172 ruler (*Oda et al., 2014a*). Besides the snake MIP, another structure inside the B-tubule (spine MIP) appears to exhibit properties required of a 48 nm molecular ruler - forming a contiguous structure, spanning 48 nm and having heterogeneities that make unique contacts to adjacent MIPs (*Figure 8—figure supplement 1*).

## Materials and methods

### Key resources

BSF single marker (BSSM) and PCF (*Wirtz et al., 1999*) *T. brucei* cells were used. The CMF22/DRC11 knockdown line is described (*Nguyen et al., 2013*).

### Preparation of demembranated flagellum skeletons for cryoET

BSF single marker (BSSM) and PCF (*Wirtz et al., 1999*) *T. brucei* cells were cultured as described (*Shimogawa et al., 2015*; *Saada et al., 2014*) and authenticated based on selective and morphogenetic markers. Cells, $2 \times 10^8$ for BSF or $4 \times 10^8$ for PCF, were washed three times in sterile 1xPBS. Supernatant was aspirated to ensure all of the PBS is removed. To remove the cell membrane and other soluble proteins and release the DNA,160 µl Extraction buffer (20 mM HEPES pH: 7.4, 1 mM $MgCl_2$, 150 mM NaCl, 0.5% NP40 IGEPAL CA-630 detergent, 2x Protease Inhibitors Cocktail-Sigma EDTA-free) + 1/10 vol 10x DNase buffer + 1/10 vol DNase (TURBO, Life Technologies 2 U/µl) was added and incubated at room temperature for 15 min. In order to solubilize the subpellicular microtubules, 1 mM $CaCl_2$ (2 µl of 100 mM $CaCl_2$) was added and incubated on ice for 30 min. Then flagellum skeletons (axoneme with PFR, basal body and FAZ filament) were centrifuged (1500 g at 4°C for 10 min) and the supernatant was removed. Then flagellum skeletons were purified away from cell body remnants and debris by one further centrifugation step over a 30% sucrose cushion at,800g at 4°C for 5 min (Extraction buffer w/o NP-40; 30% w/v sucrose). Flagellum

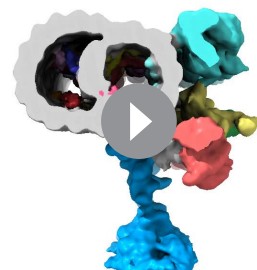

**Video 7.** 3D surface rendering of the averaged 96-nm axonemal repeat from BSF *T. brucei*, rotated to show the structures of DMT (grey), Radial spokes (blue), NDRC (green), f IC/LC (yellow), OAD (cyan) and IAD (red). Structures other than the Snake MIP (mauve) and TbMIP3 (red, yellow and orange) fade away to emphasize the Snake MIP structure and its connection to TbMIP3 substructures.

https://elifesciences.org/articles/52058#video7

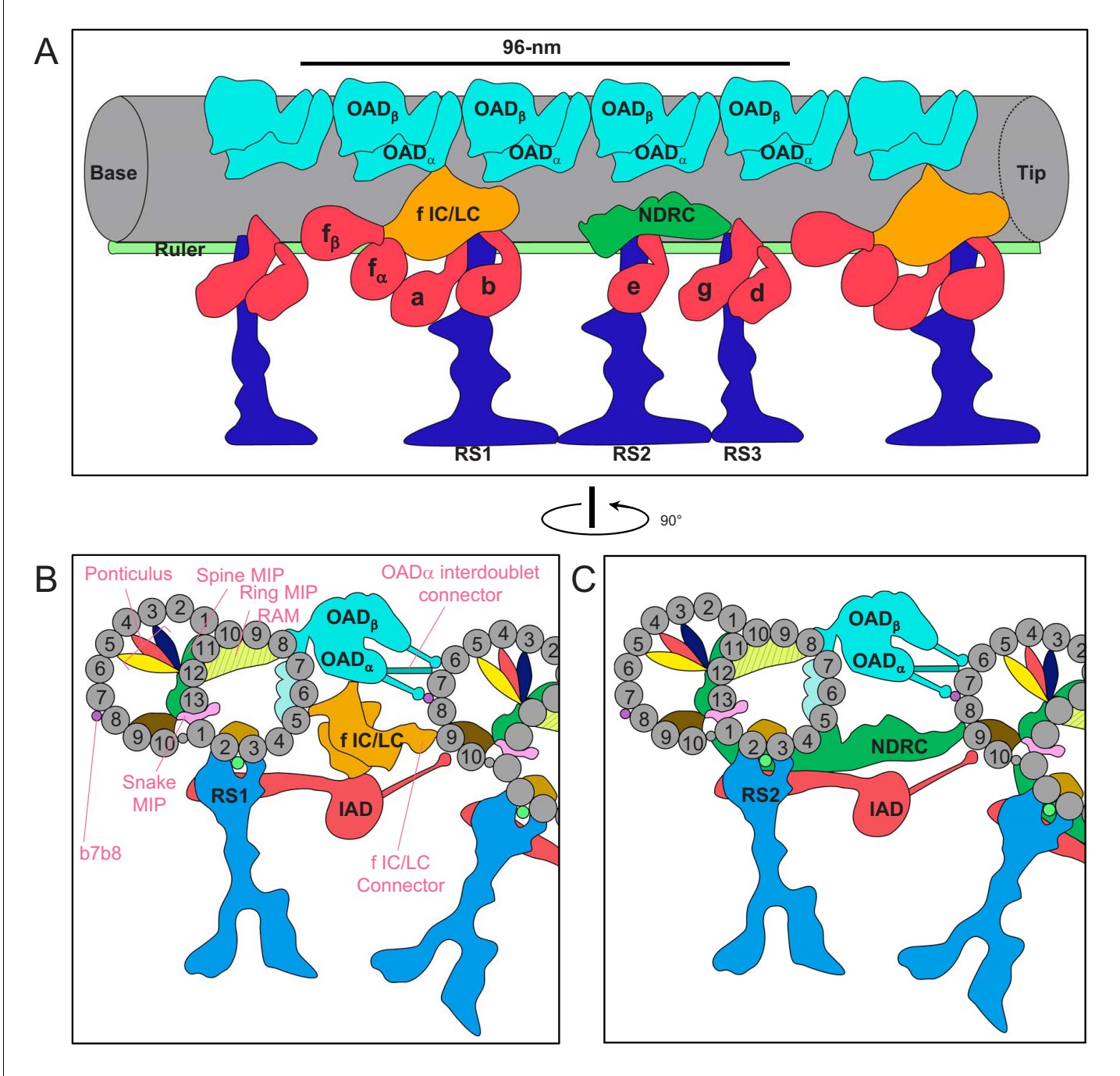

**Figure 11.** Schematic overview of the trypanosome axoneme. (**A**) Cartoon longitudinal view of the entire averaged 96-nm axonemal repeat. Major labeled structures are Outer Arm Dyneins (OAD), Inner Arm Dyneins (IAD), dynein-f IC/LC, Nexin Dynein Regulatory Complex (NDRC), Radial Spokes (RS) and Ruler. Image is oriented with proximal end (base) at the left. (**B and C**) Cartoon cross-section view of the axoneme (viewed from the proximal end) at roughly the position of RS1 (**B**) and RS2 (**C**). Protofilaments are numbered and structures are labelled as for panel A. Major trypanosome-specific structures described in text are labelled in pink. Note that additional *T. brucei*-specific structures, RingMIP, RAM MIP and b-connector are not visible in this simplified depiction. Summary of MIP structures is provided in **Supplementary file 1**.

skeletons from 200 µl of the upper fraction of the buffer-sucrose interface were collected and washed twice in 200 µl Extraction buffer, centrifugation at 1500 g at 4°C for 10 min, then resuspended in 40 µl buffer. Samples were either mixed with gold beads and plunge frozen immediately, as described below, or assessed directly for sample quality. To assess sample quality, BSF samples

were negative-stained and analyzed using an FEI T12 transmission electron microscope equipped with a Gatan 2k × 2 k CCD camera. Samples were intact with uniform length distribution and a mean length of 25.2, + /- 3.5 µm (*Figure 2B–F*). PCF samples were examined by light microscopy to ensure uniform length distribution.

## CryoET sample preparation and tilt-series acquisition

BSF or PCF samples in the amount of 40 µl was mixed with either 5 nm (for BSF) or 10 nm (for PCF) diameter fiducial gold beads in 12:1 ratio. An aliquot of 3 µl of the axoneme-gold beads solution was applied onto Quantifoil (3:1) holey carbon grids (for BSF) or continuous carbon-coated EM grids (for PCF) which were freshly glow-discharged for 30 s at −40 mA. Excess of the sample on the grid was blotted away with a filter paper, at a blot force of −4 and blot time of 5 s, and vitrified by immediately plunging into liquid nitrogen-cooled liquid ethane with an FEI Mark IV Vitrobot cryo-sample plunger. Axoneme architectural integrity and gold bead concentration were assessed and plunge-freezing conditions optimized by obtaining low-resolution cryoET tilt series in an FEI TF20 transmission electron microscope equipped with an Eagle 2K HS CCD camera. From these tilt series, cryoET tomograms were evaluated to ensure structural integrity of the axoneme and PFR. Vitrified cryoET grids were stored in liquid nitrogen until use.

For high-resolution cryoET tilt series acquisition, vitrified specimens were transferred with a cryo-holder into an FEI Titan Krios 300kV transmission electron microscope equipped with a Gatan imaging filter (GIF) and a Gatan K2 Summit direct electron detector. Samples were imaged under low-dose condition using an energy filter slit of 20 eV. CryoET tilt series were recorded with *SerialEM* (*Mastronarde, 2005*) by tilting the specimen stage from −60° to +60° with 2° increments. The cumulative electron dosage was limited to 100 ~ 110 e$^-$/Å$^2$ per tilt series. All 4k × 4 k frames were recorded on a Gatan K2 Summit direct electron detector in counting mode with the dose rate of 8–10 e$^-$/pixel/s. For each tilt angle, a movie consisting of 7 to 8 frames was recorded. For the PCF samples, the nominal magnification was x26,000, giving rise to a calibrated pixel size of 6.102 Å. The defocus value was targeted at −4 µm. When the BSF samples were ready to be imaged, the same instrument was upgraded with a VPP, allowing us to obtain higher contrast images at closer to focus and higher magnification conditions. To obtain tilt series for the BSF samples with VPP, we follow the procedures previously described (*Fukuda et al., 2015*; *Si et al., 2018*) and used the same GIF and K2 parameters as indicated above. Before starting each tilt series, we moved to a new VPP slot, waited for 2 min for stabilization, then pre-conditioned the VPP by illumination with a total electron dose of 12 nC for 60 s to achieve a phase shift of ~54°. Tilt series were recorded at a nominal magnification of 53,000X (corresponding to a calibrated pixel size of 2.553 Å) and a targeted defocus value of −0.6 µm. For BSF we collected a total of 50 tomograms and selected the 10 best, based on limited axoneme compression for sub-tomogram averaging. Cross sections of these 10 tomograms are shown in *Figure 3—figure supplement 1*, and have circularity, measured as ratio of short axis/long axis, ranging from 0.92 to 0.98. This yielded 763 particles that were averaged to determine the 3D structure of the BSF axonemal repeat. For WT PCF we collected 27 tomograms, and 17 of them were used for sub-tomogram averaging, resulting in 1177 particles averaged. For DRC11/CMF22 RNAi samples a total of 24 tomograms were collected and 19 of them were used for sub-tomogram averaging, resulting in 1726 particles averaged. For sub-tomogram averaging of individual DMT (*Figure 6* and *Figure 6—figure supplement 1*), an additional 24 tomograms of BSF axonemes were used, for a total of 34 tomograms, yielding 297 to 339 particles averaged for each DMT (DMT1 = 339, DMT2 = 332, DMT3 = 297, DMT4 = 327, DMT5 = 311, DMT6 = 337, DMT7 = 316, DMT8 = 306, DMT9 = 309).

## Data processing

For PCF and BSF samples, frames in each movie of the raw tilt series were drift-corrected, coarsely aligned and averaged with *Motioncorr* (*Li et al., 2013*), which produced a single image for each tilting angle. The tilt series images were reconstructed into 3D tomograms by weighted back projections using the *IMOD* software package (*Kremer et al., 1996*) in six steps. Micrographs in a tilt series were coarsely aligned by cross-correlation (step 1) and then finely aligned by tracking selected gold fiducial beads (step 2). The positions of each bead in all micrographs of the tilt series were fitted into a specimen-movements mathematical model, resulting in a series of predicted positions.

The mean residual error was recorded to facilitate bead tracking and poorly-modeled-bead fixing (step 3). With the boundary box reset and the tilt axis readjusted (step 4), images were realigned (step 5). Finally, tomograms were generated by weighted back projection (step 6). Contrast transfer function (CTF) was corrected with the *ctfphaseflip* program (**Xiong et al., 2009**) of IMOD in step five above. The defocus value for each micrograph was determined by *CTFTILT* (**Mindell and Grigorieff, 2003**), and the estimated defocus value was used as input for *ctfphaseflip*. Note, one of the benefits of using a phase plate is that the CTF is insensitive to the sign of the defocus value being negative (underfocus) or positive (overfocus) (**Fan et al., 2017**).

To improve the signal-to-noise ratio and enhance the resolution, sub-tomograms containing the 96-nm axonemal repeated units along each DMT were extracted/boxed out from the raw tomograms. Sub-tomogram averaging and the missing-wedge compensation were performed using *PEET* program (**Nicastro et al., 2006**; **Heumann et al., 2011**) as detailed previously (**Si et al., 2018**), except for a new script we wrote to pick sub-volumes as outlined in the subsequent paragraphs.

In our sub-tomogram averaging scheme, each particle is defined as the 96-nm repeating unit of the DMT. We developed a *MATLAB* script, *autoPicker*, to semi-automatically pick particles and calculate their location and orientation based on axoneme geometry. Briefly, we represent the 9+2 axoneme as a cylinder. For each axoneme in a tomogram, we used *IMOD* to visually pinpoint 11 points and save their coordinates into a file. The first two points, $p_a$ and $p_b$, are the center points of the two bases of the cylinder. The remaining 9 points ($p_i$, i=1...9) identify the centers of the nine DMTs (particles) within the first 96-nm length at one end of the selected axoneme. The center is defined as the intersection point of a DMT with the middle of the three radial spokes along each particle's 96-nm unit length. Our script reads the coordinates of the 11 points, calculates vector $\overrightarrow{p_a p_b}$ that defines the orientation of the cylinder, determine the center coordinates of all other particles within this axoneme based on the following formula:

$$p_{ij} = p_i - L \cdot j \cdot \frac{\overrightarrow{p_a p_b}}{\left| \overrightarrow{p_a p_b} \right|}.,$$ where i = 1, 9; j = 1 to $\left| \overrightarrow{p_a p_b} \right| / L$, L is the unit length (96nm)

In order to uniquely identify the orientation of each particle, *autoPicker* also calculates a second point, $p^*_{ij}$ for each $p_{ij}$. $p^*_{ij}$ corresponds to the middle radial spoke's end near the central pair. This is accomplished by solving the following linear algebraic equations that both $p^*_{ij}$ and $p_{ij}$ must satisfy (see illustrations in *Figure 3—figure supplement 4*):

$$\begin{cases} \overrightarrow{p_a p_b} \cdot \overrightarrow{p_{ij} p^*_{ij}} = 0 \\ \left( \overrightarrow{p_a p_b} \times \overrightarrow{p_a p_{ij}} \right) \cdot \overrightarrow{p_{ij} p^*_{ij}} = 0 \\ \left| \overrightarrow{p_{ij} p^*_{ij}} \right| = Length\ of\ the\ radial\ spoke\ (60nm) \end{cases}$$

We ran *autoPicker* for each axoneme in our tomograms to generate a *PEET* mod file that contains a list of the above described $p_{ij}$ and $p^*_{ij}$ pairs for all particles in that axoneme. Program *stalkInit* in *PEET* then read this mod file and generate an initial *motive* list file, a RotAxes file and three model files containing the coordinates for each particle. *PEET* then read the coordinate and orientation information from these files and automatically extracted the particles from the tomograms to perform iterative sub-tomogram averaging until no further improvement can be obtained.

Sub-tomogram averaging of the individual DMTs was performed in two steps. Step1: particles (96-nm repeat units), picked from all 9 DMTs were classified into nine classes, corresponding to the DMT from which each particle was picked, DMT 1–9. Step 2: for particles in each of the nine classes, sub-tomogram averaging was performed using PEET.

The resolutions of the sub-tomogram averages were evaluated by two different approaches, one based on Fourier shell correlation (FSC) calculated by *simpleFSC* in *PEET* (**Nicastro et al., 2006**; **Heumann et al., 2011**) and the other by *ResMap* (**Kucukelbir et al., 2014**). To calculate FSC curves, we split all particles into two of equal-sized subsets following the PEET tutorial. Specifically, particles are separated into two subsets with the *PEET* specific *motive list* file by designating each sub-volume as either '1' or '2' so that it would be placed into one of the two sub-sets. *PEET* then performed sub-tomogram averages independently for particles in each of the two equal-sized sub-sets, yielding two sub-tomogram averages of the 96-nm axonemal structure. These two independently calculated sub-tomogram averages were then used as the input maps of the *simpleFSC* program in the PEET

package to calculate the FSC curve for the entire 96-nm axonemal repeat (*Figure 3—figure supplement 2A*). We also calculate FSC curves for local regions encompassing DMT with MIPs, OAD, IAD, NDRC or RS. To do so, a cuboid mask was used in *ChimeraX* (*Goddard et al., 2018*) to extract two corresponding local density regions that primarily containing either DMT with MIPs, or OAD, or IAD, or NDRC or RS from the two sub-tomogram averages. Each set of two corresponding cuboid volumes (*Figure 3—figure supplement 2B*) was then used as the input maps of the *simpleFSC* program in the PEET package to calculate an FSC curve for the local region, which is plotted as a function of spatial frequency (*Figure 3—figure supplement 2A*). Local resolution across the entire averaged 96-nm axonemal repeat was also evaluated with *ResMap* (*Kucukelbir et al., 2014*) using the above two independently calculated sub-tomogram averages as input maps and the result is visualized from different views in *Figure 3—figure supplement 2C*).

## 3D visualization

*IMOD* (*Kremer et al., 1996*) was used to visualize the reconstructed tilt-series and the 2D tomographic slices of the sub-tomogram averages. UCSF *ChimeraX* (*Goddard et al., 2018*) was used to visualize the resulting sub-tomogram averages in their three dimensions. Segmentation of densities maps and surface rendering for the different components of the 96-nm repeated unit were performed by the tools *volume tracer* and *color zone* in UCSF *Chimera* (*Pettersen et al., 2004*). GIMP 2.8.18 (GNU Image Manipulation Program) was used to color regions of interest (*Figures 5*, *6B–D*, *8B–F* and *9B–C*; *Figure 3—figure supplement 3C–D*, *Figure 8—figure supplement 1B*, *Figure 9—figure supplement 1B*; *Supplementary file 1*). For rendering, no filters were applied on MIPS but we applied low pass filters on the other components to improve the clarity of individual structures described in the text. For the structures in *Figure 3C–E*; *Figure 4A,B,D*; *Figure 7A–E*, we filtered the DMT, NDRC, RS, IC/LC, OAD and IAD to 30 Å. For the structures in *Figure 5*; *Figure 6*; *Figure 6—figure supplement 1*, we filtered the entire map to 50 Å).

## Trypanosome motility videos

Motility videos of BSF cells were obtained exactly as described in *Kisalu et al. (2014)*. Motility videos of PCF cells were obtained exactly as described in *Nguyen et al. (2013)*. All videos were recorded and played back at 30 frames per second. The PCF tetracycline-inducible DRC11/CMF22 RNAi knockdown line has been described previously (*Nguyen et al., 2013*). WT and mutant PCF videos correspond to this knockdown line cultured in the absence (WT) or presence (mutant) of 1 µg/ml tetracycline to induce RNAi.

## Data availability

All data generated or analyzed during this study are included in the manuscript and supporting files. Source data files have been provided for *Figure 2F* and *Figure 3—figure supplement 4*. The cryoET sub-tomogram average maps have been deposited in the EM Data Bank under the accession codes EMD-20012, EMD-20013 and EMD-20014, for the wild-type bloodstream form, wild-type and DRC11-knock-down procyclic form, respectively.

## Acknowledgements

We thank Changlu Tao for technical assistance in SerialEM operation, Robert Minahan and Masahiro Yabe for help in data processing, Neville Kisalu and Michelle Shimogawa for motility videos of BSF cells. We thank Michelle Shimogawa for critical reading of the manuscript. This research has been supported in part by grants from NIH (R01GM071940, AI052348). SNF (P300PA_174358 and P2BEP3_162094). HK was supported by NIH-NRSA fellowship GM007185.We acknowledge the use of instruments in the Electron Imaging Center for Nanomachines supported by UCLA and grants from NIH (S10RR23057, S10OD018111 and U24GM116792) and NSF (DMR-1548924 and DBI-1338135).

## Additional information

### Funding

| Funder | Grant reference number | Author |
|---|---|---|
| Swiss National Science Foundation | P300PA_174358 | Simon Imhof |
| National Institutes of Health | R01GM071940 | Jiayan Zhang<br>Hui Wang<br>Ivo Atanasov<br>Wong H Hui<br>Z Hong Zhou |
| National Institutes of Health | S10RR23057 | Z Hong Zhou |
| National Science Foundation | DMR-1548924 | Z Hong Zhou |
| National Institutes of Health | AI052348 | Simon Imhof<br>Hoangkim Nguyen<br>Kent L Hill |
| Swiss National Science Foundation | P2BEP3_162094 | Simon Imhof |
| National Institutes of Health | S10OD018111 | Z Hong Zhou |
| National Institutes of Health | U24GM116792 | Z Hong Zhou |
| National Science Foundation | DBI-1338135 | Z Hong Zhou |

The funders had no role in study design, data collection and interpretation, or the decision to submit the work for publication.

### Author contributions

Simon Imhof, Conceptualization, Formal analysis, Funding acquisition, Methodology, Writing—original draft, Writing—review and editing; Jiayan Zhang, Conceptualization, Formal analysis, Investigation, Methodology, Writing—original draft, Writing—review and editing; Hui Wang, Khanh Huy Bui, Hoangkim Nguyen, Formal analysis, Investigation, Methodology, Writing—review and editing; Ivo Atanasov, Wong H Hui, Shun Kai Yang, Methodology, Writing—review and editing; Z Hong Zhou, Kent L Hill, Conceptualization, Resources, Formal analysis, Supervision, Funding acquisition, Writing—original draft, Project administration, Writing—review and editing

### Author ORCIDs

Simon Imhof https://orcid.org/0000-0001-7514-6811
Jiayan Zhang https://orcid.org/0000-0003-3602-1199
Hui Wang https://orcid.org/0000-0002-9922-7170
Khanh Huy Bui https://orcid.org/0000-0003-2814-9889
Z Hong Zhou https://orcid.org/0000-0002-8373-4717
Kent L Hill https://orcid.org/0000-0001-6529-1273

### Decision letter and Author response

Decision letter https://doi.org/10.7554/eLife.52058.sa1
Author response https://doi.org/10.7554/eLife.52058.sa2

## Additional files

### Supplementary files

- Supplementary file 1. List of MIPs identified in this study in BSF *T. brucei*.

- Transparent reporting form

## Data availability

All data generated or analyzed during this study are included in the manuscript and supporting files. Source data files have been provided for Figure 2F and Figure 3-Suppl. 4. The cryoET sub-tomogram average maps have been deposited in the EM Data Bank under the accession codes EMD-20012, EMD-20013 and EMD-20014, for the wild-type bloodstream form, wild-type and DRC11-knockdown procyclic form, respectively.

The following datasets were generated:

| Author(s) | Year | Dataset title | Dataset URL | Database and Identifier |
|---|---|---|---|---|
| Imhof S, Zhang J, Wang H, Bui KH, Nguyen H, Atanosov I, Hui WH, Yang SK, Zhou ZH, Hill KL | 2019 | T. Brucei Axoneme in DRC11-knockdown procyclic form revealed by cryo electron tomography with Volta phase plate. | https://www.ebi.ac.uk/pdbe/entry/emdb/EMD-20014 | Electron Microscopy Data Bank, EMD-200 14 |
| Imhof S, Zhang J, Wang H, Bui KH, Nguyen H, Atanosov I, Hui WH, Yang SK, Zhou ZH, Hill KL | 2019 | T. Brucei Axoneme in bloodstream form revealed by cryo electron tomography with Volta phase plate. | https://www.ebi.ac.uk/pdbe/entry/emdb/EMD-20012 | Electron Microscopy Data Bank, EMD-200 12 |
| Imhof S, Zhang J, Wang H, Bui KH, Nguyen H, Atanosov I, Hui WH, Yang SK, Zhou ZH, Hill KL | 2019 | T.brucei Axoneme in procyclic form revealed by cryo electron tomography with Volta phase plate. | https://www.ebi.ac.uk/pdbe/entry/emdb/EMD-20013 | Electron Microscopy Data Bank, EMD-200 13 |

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
