## [Decision Letter]

**Acceptance summary:**

This study presents the first structures of the Trypanosome axoneme. Although cilia/flagella are found throughout evolution, structures are only available for a select few species. This structure from T. brucei is the first from the excavata supergroup, providing an important evolutionary comparison. Novel inter-doublet connections are observed beside the known Nexin Dynein Regulatory Complex (NDRC): connection to the IC/LC complex and a protrusion from the dynein linker. The NDRC connection is abolished in a CMF22/DRC11 RNAi knockdown, which shows this subunit is located close to the B-tubule and affects motility. Within the doublets, some T brucei MIP densities correspond to MIPs known from the T. thermophilus and C. reinhardtii structures. However, it also contains a large number of unidentified MIP densities which appear to be T brucei specific.

**Decision letter after peer review:**

[Editors’ note: a previous version of this study was rejected after peer review, but the authors submitted for reconsideration. The first decision letter after peer review is shown below.]

Thank you for submitting your work entitled "Cryo electron tomography reveals novel structural foundations of axoneme motility in the pathogen *Trypanosoma brucei*" for consideration by *eLife*. Your article has been reviewed by a Senior Editor, a Reviewing Editor, and four reviewers. The following individuals involved in review of your submission have agreed to reveal their identity: Benjamin D Engel (Reviewer #3); Brooke Morriswood (Reviewer #4).

Our decision has been reached after consultation between the reviewers and I regret to inform you that we felt the work could not be accepted in its current form. We felt that your manuscript's main message (the original aspects of the trypanosome axoneme) is potentially of interest to *eLife* readers, especially given the paucity of organisms generally studied in cilia biology. The reviewers also agreed that your paper brings interesting and novel insights. However, there was a strong consensus that your manuscript needs more work than would be reasonable to expect in a two month turn-around and hence our decision that it needs to be rejected at this stage. The reviewers comments are attached in full below. The main revisions required to make the paper suitable for *eLife* would be:

1) Address structural heterogeneity of the different doublet microtubules. Given the inherent asymmetric structure of the Trypanosome flagellum, with the paraflagellar rod at one side, this must be explored. You are using a small dataset and so additional tomograms may be required to resolve the different doublets.

2) Address the structure of the PFR, and thereby build on your and others prior observations of this accessory structure. Ideally you would also compare distal and proximal portions of the axoneme to account for known molecular differences (e.g. there is no PFR in the first micrometre of the axoneme).

3) Incorporate comparative images to show differences to the 96nm repeats from other organisms.

4) Redo your resolution estimations (as described by reviewer 3).

5) Overhaul your manuscript to address the reviewers comments. In particular: remove all statements about the "greatest" resolution, separate speculations from results, and include a proper summary of extant data on the 96nm repeat to place the results in proper context.

Reviewer #1:

The ciliary cytoskeleton (the axoneme) consists of nine microtubule doublets (MTDs) arranged in a nine-fold symmetric array. These doublets contain many accessory proteins such as Dyneins, Radial Spokes and microtubule inner proteins (MIPs) that are responsible for cilia motility and/or MTD stability. The proteins that are stably associated with the MTDs are recurring in regular repeats along the axoneme its length. Many efforts are and have been made to solve the structure of the axonemal repeats and inner proteins to understand cilia stability and the waveform and regulation of cilia motility.

The authors present the structure of the Trypanosoma brucei axoneme as a first representative of the excavata super group. Novel inter-doublet connections are observed beside the known Nexin Dynein Regulatory Complex (NDRC): connection to the IC/LC complex and a protrusion from the dynein linker. The NDRC connection is abolished in a CMF22/DRC11 RNAi knockdown, which shows this subunit is located close to the B-tubule and affects motility. Within the doublets, some T brucei MIP densities correspond to MIPs known from the T. thermophilus and C. reinhardtii structures. However, it also contains a large number of unidentified MIP densities which appear to be T brucei specific.

A number of axoneme structures have been solved recently. Our feeling is that there are some novel aspects to this T. brucei structure, but at the moment the findings are very descriptive. For example showing, but not identifying, the MIP densities. The authors list the differences in the 96-nm repeat, compared to other axonemal structures, but can't explain how these features relate to the function of the T.brucie axoneme.

Verifying the findings such as the other inter-doublet connections in vivo or identifying some of the MIP densities would make the manuscript more suitable for publication in *eLife*.

Reviewer #2:

This manuscript reports a high-quality data set on the 3D organization of the so-called 96-nm repeat structures in two developmental stages of the protist Trypanosoma brucei. Flagella were purified and analysed by cryo electron tomography and sub-tomogram averaging after plunge freezing. The resolution reached remarkable values (12-15 Å) for the bloodstream stage. In some cases, tubulin monomers could be resolved at the level of protofilaments. It revealed common features with other axonemes, especially from human cilia but also unique differences. Nice structural details of the internal composition of microtubules are reported, including the unique ponticulus and the discovery of novel microtubule internal proteins (MIP). Comparison with flagella of cells where the expression of a component of the nexin-dynein regulatory complex was knocked down revealed structural modifications. The proximal lobe of the NDRC complex appeared affected, what impaired the connection with the neighbouring doublet and could explain the motility defect.

This is a very nice study, with some new developments (MATLAB script) and will be of interest for cilia biologists and structural biologists at large.

Three points need to be clarified to fully understand the results:

1) Exponentially growing cultures of trypanosomes contain about 50% of cells with two flagella (the mature one and the growing one). Based on length distribution of bloodstream form flagella (Figure 2), it seems that mostly full-length flagella are present in the sample. How is it possible? Is it due to the purification procedure that would select long flagella? This is important because at least the distal end of growing and mature flagella are known to be different in terms of structure (Höög et al., 2014) or composition (Subota et al., 2014). Please clarify. Length measurements for procyclic form flagella are missing.

2) A recent paper (Edwards et al., 2018) showed that the docking of outer dynein arms was different along the length of the trypanosome flagellum (in PCF), with specific proximal and distal docking complexes. Subsection “3D Structure of the trypanosome 96-nm axonemal repeat”: "tilt series were collected from the center part of the flagellum, spanning the middle third between the basal body and tip". A cartoon would help to avoid ambiguity but if we understood correctly, it seems that the central portion of the flagellum was used. It is where the two docking systems are likely to overlap, hence potentially generating heterogeneity. Is the organisation the same in the short portion without paraflagellar rod (PFR)?

3) Which doublets (out of the 9) were selected for analysis? The same group has shown that doublets were not equivalent (Hughes et al., 2012), especially those connected to the PFR where dynein arms looked different.

The Title and the interpretation should be tuned down, although unique structural features (especially MIPs) are indeed reported, there is little direct evidence that they contribute to the original axoneme motility.

Writing-up. There is a lot of interpretation in the result section. The authors should either remove these and do a more exhaustive Discussion section, or write the paper with Results section and Discussion section combined.

Reviewer #3:

This study by Imhof et al., presents the first structures of the Trypanosome axoneme. Although cilia/flagella are found throughout evolution, structures are only available for a select few species. This structure from T. brucei is the first from the excavata supergroup, providing an important evolutionary comparison. The authors present several interesting findings, including the descriptions of only two dynein motors per OAD, increased connections between doublet microtubules (DMTs), and several lineage-specific microtubule inner proteins (MIPs). They also use an RNAi knockdown line to determine the position of DRC11. In principle, I support publication of this work, and I think the data is a valuable addition to the axoneme field. However, there are several major points related to the analysis that must be addressed, in particular related to resolution estimation and exploring structural variation between different DMTs. In addition, there is too much pure speculation about the potential functions of several structures, with no experimental evidence to support these functions. This speculation should be removed or heavily qualified.

Essential revisions:

1) The authors heavily promote the resolution of their structure, claiming "This resolution is the highest cryoET structure yet reported for the 96-nm repeat from any organism". However, there are some serious issues with the resolution estimation, and as a result, I believe the resolution has been overstated.

The paper's abstract claims that the DMT is resolved to 12Å. This is based on the ResMap analysis shown in Supplemental Figure S3. However, this analysis is troubling. First of all, the resolutions on this map appear to primarily range from 12-16Å, whereas the FSC curve shows a global resolution of 21Å at the 0.143 cutoff. If all the local resolutions from ResMap are averaged, the result should be close to 21Å. However, it does not appear that this will be the case, and instead ResMap is estimating resolutions that are at least 5Å better than the FSC. Even the less well-resolved appendage structures that can be seen in this image range from 15-18Å. This is a bit difficult to judge from the figure because the authors have intentionally only shown the backside of the DMT (which reports the highest resolution). They must also show the other side, with all the important accessory structures (similar to Figure 3D), as well as a cross-section slice through the DMT to show the MIPs (similar to Figure 3B). Only then can we see the range of resolutions estimated by ResMap. But even in the view that is shown, the appearance on the DMT is way too speckled, with a huge dynamic range of 12-16Å on the microtubule wall. This noisy surface is a clear sign that something is wrong with the ResMap-the surfaces should look much smoother, with less hotspots of resolution variation.

As the ResMap cannot be taken at face-value, a parallel approach should be attempted to estimate local resolution. I recommend using masks to perform two local FSCs – the doublet region and the region containing the appendage structures (OAD, IAD, RS, NRDC). How do these FSCs compare to the global FSC and to the ResMap?

For the isosurface renderings throughout the paper, the DMT looks properly filtered, but the appendage structures appear to be oversharpened or displayed at a resolution that is too high (their surfaces look noisy). I assume that these maps were uniformly filtered to the same resolution. Was it the global 21Å? I can't seem to find this information in the paper. I would expect that by around 20Å, the holes in the middle of the dynein AAA+ rings would start to become visible, at least as indentations. But in these maps, the AAA+ rings look like round egg-shaped blobs, another sign that the resolution is not as high as claimed.

The authors used a tilt-series acquisition scheme that starts at -60 degrees and thus destroys the high-resolution information before reaching low tilt, as opposed to the much preferred dose-symmetric scheme starting at zero degrees combined with dose-weighting (see the high-resolution HIV work by John Briggs and Wim Hagen). Furthermore, there currently is no way to correct the contrast transfer function (CTF) for low dose cryo-tilt series acquired with the Volta Phase Plate. The authors thus did not perform CTF-correction, meaning that resolution of the average is limited to the first zero of the power spectrum. Given the -0.6 μm target defocus and the large defocus gradient that is present in thick samples such as these axonemes, especially at higher tilts, I anticipate that this first zero would strongly limit the resolution (20-25Å sounds about right, not 12Å). Therefore, I am very cautious of the bold resolution claims made in this paper.

Finally, the authors use a "gold-standard" FSC to determine resolution, but it was not clear to me from the methods when exactly the extracted particles were split into two half-sets and averaged independently, as is required for gold-standard assessment. With only 700 total particles, getting two half-sets to 21Å might be challenging. Please explicitly describe how the averaging was performed instead of just "as described previously (75) using PEET".

2) The focus of this paper is to show the evolutionary differences of the T. brucei axoneme. Of course, by far the most distinct feature of trypanosome flagella is the paraflagellar rod (PRF). This structure seems important for axoneme stability under strong forces, a key question the authors sought to address. In Figure 2G, the authors show clear periodic connections between the axoneme and PFR (marked with arrows). It would be very valuable to compare averages of PFR-linked versus non-PRF-linked DMTs. This should be a relatively easy task, just splitting particles into those two categories based on their location with respect to the PFR. Furthermore, the PFR appears to have a fairly regular structure in Figure 2G, so is it possible to generate an average of the PFR itself? Such a structure would be something really new, and would add value to this paper.

3) The globular density with an 8-nm repeat in Figure 3—figure supplement 3 has a weak signal, and thus appears to have low occupancy in this average. Why is this? Might it have something to do with the connection to the PFR? Judging by Figure 3A, one would expect about a third or a fourth of the DMTs to have a connection to the PFR. The authors should investigate how the occupancy of this density varies between DMTs by producing averages of different DMTs using their radial position relative to the PFR for reference. Without more information, the author's proposed function of this density in regulating dynein binding (subsection “3D Structure of the trypanosome 96-nm axonemal repeat”) is far too speculative.

4) Based on the strong signal of the IAD-f IC/LC density, the authors conclude: "An IAD-f IC/LC interdoublet connection is observed between three specific doublet pairs in *Chlamydomonas* (17). However, the prominence of the IAD-f IC/LC connection in T. brucei suggests it is present between most and perhaps all doublet microtubules, indicating that nexin links in T. brucei include both the NDRC and IAD-f IC/LC. This distinguishes T. brucei axonemes from the known 3D axoneme structures from other organisms (36, 46)." Similar to points 2 and 3 above, if the authors want to make this claim, then they absolutely must examine the density in averages of different DMTs around the T. brucei axoneme.

5) The issues raised in the four points above (problems with resolution estimation and neglecting to analyze structural variation between different DMTs) are all related to the very limited dataset used in this paper. The primary "high resolution" structure in this paper (wild-type BSF) was generated from only 700 particles from 10 tomograms. This is only half a day of acquisition on a Titan Krios microscope (tomograms take about 1 hour each). While I understand it is not *eLife* policy to ask for more experiments, I think it is completely appropriate in this case for the authors to spend one more day on the microscope with their already prepared cryo-EM samples to acquire 20 more tomograms. This would produce a sufficient dataset to perform classification and look at how specific densities vary between different DMTs (see points 2-4 above). I understand that the Volta Phase Plate used in this study enables averages to be generated with less particles, but I think the 700 particles in the current average are too few to properly do this analysis, and it is not clear to me why the authors chose to proceed with such a limited dataset.

6) The proximal and distal holes in the DMTs look convincing. However, what is not convincing to me is their proposed function of allowing MIPs to be incorporated after completion of the DMT. The holes are tiny, only about as wide as a tubulin monomer (4 nm), and thus do not seem big enough to allow the free transit of MIPs, which are significantly larger than the holes. Perhaps the holes could serve as a location for the start of an "unzipping" event between the A- and B-tubules, which could allow insertion of larger MIP structures such as the ponticulus, but this is completely speculative. I don't think much can be said about the function of the DMT holes at this point.

7) The extended discussion of the RingMIP function is highly speculative and should perhaps be omitted or at least down-weighted. Its proposed mechanosensory role is not proven by the data in this paper, just speculated.

8) The authors should be careful with stating the significance of the comparison between developmental stages; there's no clear conclusion from this comparison (other than a low-resolution hint of MIPA3-4). So, putting this in the abstract without indicating the negative result could be considered false advertising. Upon reading the abstract, I assumed that there were developmental differences, and I was disappointed when I finally discovered at the end of the paper that there were not.

Reviewer #4:

Imhof, Zhang et al., present cryo-electron tomography data on preparations of isolated axonemes from the unicellular parasite Trypanosoma brucei, eukaryotic supergroup Excavata. Specifically, they provide a detailed analysis of the 96nm repeat that forms the core structural unit of the axoneme. These data were obtained from two life cycle stages of the organism – the slender bloodstream form which is found in infected mammalian hosts and whose motility is better characterised, and also the procyclic form which is found in the midgut of the tsetse fly vector. Not only are the usual structural features such as the outer dynein arms, radial spokes resolved, but also some fascinating observations of the microtubule inner proteins are provided, which appear to be considerably more abundant in the axonemes of this organism than in others imaged to date.

Given that the axoneme is an almost ubiquitous structure that was present in the last eukaryotic common ancestor, the data here are extremely significant and of relevance far beyond the trypanosome and parasitology community. As the authors note, taxonomic sampling of the ultrastructure of the 96nm axonemal repeat is limited, and by providing data from the supergroup Excavata the authors have considerably broadened the perspective onto this structure. The observations of the microtubule inner protein complexes are fascinating and could set the stage for considerable future work.

It is important to note that I have no first-hand practical experience of cryo-electron tomography and am therefore not qualified to judge the technical aspects of the manuscript relating to this technique. I am happy for the opinions of the other referee(s) to take precedence on this point.

The data appear to be of very high quality and the figures do not require more than cosmetic alteration. No extra practical work seems required, I think, and I have no major concerns. The manuscript could be considerably tightened in order to do full justice to the quality of the results, however.

In particular, the Introduction needs much more detail. The authors should define the 96nm axonemal repeat and explicitly summarise previous work on this structural unit. This is important for placing the results in context and defining the paper's original contribution. The 96nm repeat is currently first mentioned in the Results section without any prior introduction. There has been substantial previous work on the morphology and ultrastructure of the trypanosome axoneme, so it is important to emphasise that this work focuses on the structure of the axoneme's 96nm repeat unit specifically. Presently there is considerable potential for being misread, and means that the authors are actually underselling what they have.

The manuscript also frequently rushes over the results. Panels 2A-2G are not individually cited, nor are 3A-3E, and this trend continues with the other figures. Ensure that all panels are cited, and ideally in figure order (i.e. A-B-C etc) for clarity. Figure 10 is currently not cited at all in the manuscript text. Not all the supplemental figures and movies appear to be cited in the manuscript text – this needs checking. The section on the comparison of the different developmental stages (currently Figure 9) could perhaps be moved so that it comes after Figure 4.

The text would benefit from some proofreading for English, and also for style (e.g. Results should always be presented in past tense, but the present is often used here). Check also that all panel citations are correct – there is some mix-up, particular of Figure 5 and Figure 6. There is also perhaps a bit too much interpretation in the Results section that would fit better in the Discussion section.

[Editors’ note: what now follows is the decision letter after the authors submitted for further consideration.]

Thank you for submitting your article "Cryo electron tomography reveals novel structural foundations of the 96-nm axonemal repeat in *Trypanosoma brucei*" for consideration by *eLife*. Your article has been reviewed by John Kuriyan as the Senior Editor, a Reviewing Editor, and three reviewers. The following individuals involved in review of your submission have agreed to reveal their identity: Benjamin D Engel (Reviewer #1); Brooke Morriswood (Reviewer #3).

The reviewers have discussed the reviews with one another and agreed that you have done an admirable job adding significant new data and analysis to this study, including new tomograms, doublet-specific subtomogram averaging, structural comparisons between species, and much more believable resolution estimates. The manuscript is now well accessible to a general reader. All our concerns are addressed, and the manuscript requires only minor revisions. Please aim to submit the revised version within two months.

Summary:

This manuscript reports a high-quality data set on the 3D organization of the so-called 96-nm repeat structures in two developmental stages of the protist Trypanosoma brucei. This is the first representative axoneme structure of the excavata super group. Flagella were purified and analysed by cryo electron tomography and sub-tomogram averaging after plunge freezing. The resolution reached remarkable values (12-15 Å) for the bloodstream stage. In some cases, tubulin monomers could be resolved at the level of protofilaments. It revealed common features with other axonemes, especially from human cilia but also unique differences. Nice structural details of the internal composition of microtubules are reported, including the unique ponticulus and the discovery of novel microtubule internal proteins (MIP). Comparison with flagella of cells where the expression of a component of the nexin-dynein regulatory complex was knocked down revealed structural modifications.

---

## [Author Response]

Our decision has been reached after consultation between the reviewers and I regret to inform you that we felt the work could not be accepted in its current form. We felt that your manuscript's main message (the original aspects of the trypanosome axoneme) is potentially of interest to eLife readers, especially given the paucity of organisms generally studied in cilia biology. The reviewers also agreed that your paper brings interesting and novel insights. However, there was a strong consensus that your manuscript needs more work than would be reasonable to expect in a two month turn-around and hence our decision that it needs to be rejected at this stage. The reviewers comments are attached in full below. The main revisions required to make the paper suitable for ELife would be:1) Address structural heterogeneity of the different doublet microtubules. Given the inherent asymmetric structure of the Trypanosome flagellum, with the paraflagellar rod at one side, this must be explored. You are using a small dataset and so additional tomograms may be required to resolve the different doublets.

As requested, we have now completed additional experiments to examine heterogeneity among doublet microtubules, including analysis of 24 additional tomograms, as specified in the Materials and methods section of the revised manuscript. The presence of the PFR restricts possible orientations of the sample such that the axoneme is always positioned with either DMT3 or DMT7 facing the EM grid. This leads to the missing wedge problem for any individual doublet and is why we had restricted analysis to average of all doublets in our original submission. However, we recognize the importance of understanding heterogeneity, and in response to this request we have analyzed additional tomograms (now 34 total for BSF samples) and obtained structures for individual doublets. The analysis revealed new features on specific doublets and results are described in the revised manuscript (subsection “Doublet-specific features of the 96-nm repeat”, Figure 6 and Figure 6—figure supplement 1).

2) Address the structure of the PFR, and thereby build on your and others prior observations of this accessory structure. Ideally you would also compare distal and proximal portions of the axoneme to account for known molecular differences (e.g. there is no PFR in the first micrometre of the axoneme).

As requested by reviewers, we have compared PFR‐attached versus not‐PFR‐attached DMTs. As described in the text (subsection “Doublet-specific features of the 96-nm repeat”), this comparison demonstrated that PFR‐attachment does not correlate with alterations in the structure of the axonemal 96‐nm repeat. Thus, although there are DMT specific variations to the 96‐nm repeat (Figure 6 and Figure 6—figure supplement 1), these are not correlated with PFR attachment. Note that the PFR and PFR‐attachment complexes themselves have a 56‐nm repeat period, so would not be resolved in our analysis here, which is focused on the 96‐nm repeat.

We made a substantial effort to determine axoneme structure specifically in the proximal‐most region of the axoneme that does not contain PFR. However, this region was severely twisted (see reviewer Author response image 1 and author response video 1), thus preventing us from obtaining a high‐resolution structure. Severe twisting was not observed for the remainder of the axoneme and it is not presently known whether twisting reflects inherent features of the proximal region, or if it is imposed during sample preparation. The proximal axoneme is near the basal body/pro‐basal body complex, as well as four microtubules that extend from the basal body and wrap in a helix around the axoneme (Lacomble et al., 2009). These additional structures are expected to restrict sample orientation and if that orientation differs from the orientation allowed by the PFR, it would cause the proximal region to be twisted relative to the remainder of the axoneme. Future work will aim to address this issue. We have clarified in the text (subsection “3D Structure of the trypanosome 96-nm axonemal repeat”, Figure 2B) that the 96‐nm repeat structure presented represents the middle third of the axoneme.

Regarding the PFR itself, the PFR is a massive structure, with a repeat period (56‐nm) (Koyfman et al., 2011) that differs from that of the axonemal 96‐nm repeat period. Determining high‐resolution, 3D structure of the PFR is a focus of ongoing and future efforts, but is beyond the scope of the current work, which is focused on the axonemal 96‐nm repeat.

**Author response image 1. respfig1:** Proximal end of the axoneme is severely twisted. 1‐Å‐thick digital slices from a representative tomogram of the BSF axoneme near the basal body (BB) region. Two different sections, separated by 12 nm, are shown (120 and 132), with example twisted regions of the DMTs indicated by the red arrows.

**Author response video 1. respvideo1:** Proximal end of the axoneme is severely twisted. Video of slices through the tomogram presented in Reviewer Figure R1, indicating twisting of the axoneme near the basal body. Video is compressed using H.265/HEVC MP4 video format.

3) Incorporate comparative images to show differences to the 96nm repeats from other organisms.

As requested, we have added images of 96‐nm repeats from other organisms (Figure 5). We selected organisms representing the eukaryotic supergroups discussed in Figure 1, including two within SAR, *Chlamydomonas* and *Tetrahymena*, as these represent two substantially divergent lineages in this supergroup. We also include a cartoon (Figure 11) that highlights distinctive and trypanosome‐specific structures discovered in the current work. Citations are provided for readers to refer to prior publications of the other organisms for detailed description of the 96‐nm repeat in each.

4) Redo your resolution estimations (as described by reviewer 3).We have done this as requested. (See also, our detailed responses to reviewer 3.)5) Overhaul your manuscript to address the reviewers comments. In particular: remove all statements about the "greatest" resolution, separate speculations from results, and include a proper summary of extant data on the 96nm repeat to place the results in proper context.

We have overhauled the manuscript text as requested. The entire text has been substantially changed, so individual changes are not specifically highlighted in the text, but lines relevant to specific reviewer and editor comments are indicated in this response letter. Statements of ‘greatest’ resolution are removed, and Results are separated from speculations. To place the results in proper context, a summary of extant knowledge regarding the 96‐nm repeat is provided in the Introduction, Figure 5 is added, and a cartoon is provided (Figure 11) to highlight relevant structures discovered here that are distinctive to *T. brucei.*

Reviewer #1:The ciliary cytoskeleton (the axoneme) consists of nine microtubule doublets (MTDs) arranged in a nine-fold symmetric array. These doublets contain many accessory proteins such as Dyneins, Radial Spokes and microtubule inner proteins (MIPs) that are responsible for cilia motility and/or MTD stability. The proteins that are stably associated with the MTDs are recurring in regular repeats along the axoneme its length. Many efforts are and have been made to solve the structure of the axonemal repeats and inner proteins to understand cilia stability and the waveform and regulation of cilia motility.The authors present the structure of the Trypanosoma brucei axoneme as a first representative of the excavata super group. Novel inter-doublet connections are observed beside the known Nexin Dynein Regulatory Complex (NDRC): connection to the IC/LC complex and a protrusion from the dynein linker. The NDRC connection is abolished in a CMF22/DRC11 RNAi knockdown, which shows this subunit is located close to the B-tubule and affects motility. Within the doublets, some *T brucei* MIP densities correspond to MIPs known from the *T. thermophilus* and *C. reinhardtii* structures. However, it also contains a large number of unidentified MIP densities which appear to be *T brucei* specific.A number of axoneme structures have been solved recently. Our feeling is that there are some novel aspects to this *T. brucei* structure, but at the moment the findings are very descriptive. For example showing, but not identifying, the MIP densities. The authors list the differences in the 96-nm repeat, compared to other axonemal structures, but can't explain how these features relate to the function of the T.brucie axoneme.

*Verifying the findings such as the other inter-doublet connections* in vivo *or identifying some of the MIP densities would make the manuscript more suitable for publication in ELife.*

We appreciate the reviewer comments and concise summary. Regarding the request to identify proteins that make up novel MIPs and inter‐doublet connections: Densities within the microtubule lumen were reported more than fifty years ago (Vickerman, 1969; Anderson and Ellis, 1965) and MIPs as a ubiquitous entity of cilia were described thirteen years ago (Nicastro et al., 2006), but identities of MIP proteins were determined only recently (Stoddard et al., 2018; Ichikawa and Bui, 2018; Owa et al., 2019). Likewise, nexin links were observed in the 1960s (Gibbons, 1963), but molecular identities of these inter‐doublet linkages were not determined until recently (Lin et al., 2011). Thus, while we recognize the interest in identifying proteins that comprise novel structures just discovered in the *T. brucei* axoneme, we consider this to be a goal of future studies and beyond the scope of the current manuscript.

Regarding the concern about being descriptive: We recognize that defining structure is by nature descriptive and emphasize that our studies have identified several novel structures not previously observed in any axoneme, despite many years of prior study. We’ve tried to better articulate the importance of our findings and how they relate to trypanosome motility in the Discussion section. Moreover, a major contribution of our work is the discovery of extensive inter‐doublet connections in *T. brucei* and we go further, with functional analysis combining cryoET analysis plus RNAi knockdown, to reveal that the motility defect of CMF22/DRC11 knockdowns (Nguyen et al., 2013) is due to specific loss of the inter‐doublet connection component of NDRC, without affecting dyneins. This result is important, as dynein-independent connection between adjacent DMTs is a founding principle of the sliding filament model for axoneme motility (Satir, 1968; Viswanadha, Sale and Porter 1943; Holwill and Satir, 1990), yet direct tests of this idea have been limited because most NDRC mutants also disrupt dyneins. Moreover, the CMF22/DRC11 knockdown analysis that we conducted here has identified structural defects of NDRC mutants that were not previously known. We have revised the text of the manuscript (Discussion section) to clarify these points.

Regarding the goal of relating structures identified to function of the *T. brucei* axoneme: A critical requirement of the trypanosome axoneme is to maintain integrity in the face of physical strain imposed on the organism due to lateral attachment to the PFR and cell body, and its particular motility mechanism – both being features that are unique to trypanosomes. Two main discoveries in the current work are extensive MIPs and inter‐doublet connections that are likewise unique to trypanosomes. The complete repertoire of functions for MIPs and inter‐doublet connections remain to be determined for any organism. However, in the few cases where data is available, MIPs and inter‐doublet connections provide axoneme stability and modulate axoneme motility (Stoddard et al., 2018; Ichikawa and Bui, 2018; Owa et al., 2019; Bower et al., 2018; Bower et al., 2013). Therefore, we consider it reasonable to suggest that lineage‐specific MIPs and inter‐doublet connections in trypanosomes provide a potential explanation for how the organism maintains axoneme integrity in the face of unique physical constraints imposed by lineage‐specific flagellum architecture and motility. We have modified the text (Discussion section) to make this link between structure and function more clear.

Reviewer #2:This manuscript reports a high-quality data set on the 3D organization of the so-called 96-nm repeat structures in two developmental stages of the protist Trypanosoma brucei. Flagella were purified and analysed by cryo electron tomography and sub-tomogram averaging after plunge freezing. The resolution reached remarkable values (12-15 Å) for the bloodstream stage. In some cases, tubulin monomers could be resolved at the level of protofilaments. It revealed common features with other axonemes, especially from human cilia but also unique differences. Nice structural details of the internal composition of microtubules are reported, including the unique ponticulus and the discovery of novel microtubule internal proteins (MIP). Comparison with flagella of cells where the expression of a component of the nexin-dynein regulatory complex was knocked down revealed structural modifications. The proximal lobe of the NDRC complex appeared affected, what impaired the connection with the neighbouring doublet and could explain the motility defect.This is a very nice study, with some new developments (MATLAB script) and will be of interest for cilia biologists and structural biologists at large.

We appreciate the positive comments regarding high quality of the data and recognition of the value of the contributions to a diverse research community.

Three points need to be clarified to fully understand the results:1) Exponentially growing cultures of trypanosomes contain about 50% of cells with two flagella (the mature one and the growing one). Based on length distribution of bloodstream form flagella (Figure 2), it seems that mostly full-length flagella are present in the sample. How is it possible? Is it due to the purification procedure that would select long flagella? This is important because at least the distal end of growing and mature flagella are known to be different in terms of structure (Höög et al., 2014) or composition (Subota et al., 2014). Please clarify. Length measurements for procyclic form flagella are missing.

Thank you for pointing this out. The samples do appear to be somewhat enriched for full‐length flagella, although if 50% of the population were to have a mature flagellum and 50% have mature plus a growing flagellum, that would yield ~1/3 growing flagella, some of which may be quite close to full length. In that case, the distribution observed is not far from expected. It’s possible that the purification procedure preferentially yields long flagella, though we haven’t directly assessed that. In our experience, growing flagella tend to stay connected to the mature flagellum and if connection and overlap between filaments is too great, one cannot reliably trace each single flagellum filament, so it is possible that among the filaments that can be reliably measured, there are more mature flagella than growing flagella. In any case, the samples contain intact flagella with good preservation of structure. Regarding the variation at the distal end, this is why we avoided the distal end of the flagella for structure determination here. For PCF samples, length of flagella was not measured, but visual inspection indicated fairly uniform length distribution.

2) A recent paper (Edwards et al., 2018) showed that the docking of outer dynein arms was different along the length of the trypanosome flagellum (in PCF), with specific proximal and distal docking complexes. Subsection “3D Structure of the trypanosome 96-nm axonemal repeat”: "tilt series were collected from the center part of the flagellum, spanning the middle third between the basal body and tip". A cartoon would help to avoid ambiguity but if we understood correctly, it seems that the central portion of the flagellum was used. It is where the two docking systems are likely to overlap, hence potentially generating heterogeneity. Is the organisation the same in the short portion without paraflagellar rod (PFR)?

As requested, we’ve indicated in Figure 2 the approximate region imaged. As detailed above (response to Editor comment #2 above), we were unable to obtain a high‐resolution structure for the proximal‐most portion of the axoneme without the PFR due to extensive twisting in this region. We recognize that our analysis does not resolve differences proximal to distal in the axoneme, which is why we specify that the middle third of the axoneme is imaged.

3) Which doublets (out of the 9) were selected for analysis? The same group has shown that doublets were not equivalent (Hughes et al., 2012), especially those connected to the PFR where dynein arms looked different.

All nine doublets were averaged for the overall structure of BSF and PCF samples. Comparisons of the 96‐nm repeat in individual DMTs and PFR‐attached versus not‐attached DMTs has now been done as requested by reviewers and results described in the text (subsection “Doublet-specific features of the 96-nm repeat”). We do not see a change in the 96‐nm repeating unit that correlates specifically with presence or absence of PFR. (See also author response to Editor comment 1 and 2.) Note that in the earlier paper referred to, Hughes et al., 2012, neither the dyneins nor doublets were reported to be different, per se, but filaments from the PFR could be seen to contact dyneins. Periodicity of PFR attachment is 56 nm (Koyfman at al., 2011; Hughes et al., 2012) and this differs from the 96‐nm repeat used for averaging. Therefore, attachment complexes would not be visible in our structure, which averages a 96‐nm unit.

The Title and the interpretation should be tuned down, although unique structural features (especially MIPs) are indeed reported, there is little direct evidence that they contribute to the original axoneme motility.

As requested, we have adjusted the Title, toned down interpretations and transferred speculation to Discussion section.

Writing-up. There is a lot of interpretation in the Result section. The authors should either remove these and do a more exhaustive Discussion section, or write the paper with Results section and Discussion section combined.

As requested, we have moved interpretation into the Discussion section.

Reviewer #3:This study by Imhof et al., presents the first structures of the Trypanosome axoneme. Although cilia/flagella are found throughout evolution, structures are only available for a select few species. This structure from T. brucei is the first from the excavata supergroup, providing an important evolutionary comparison. The authors present several interesting findings, including the descriptions of only two dynein motors per OAD, increased connections between doublet microtubules (DMTs), and several lineage-specific microtubule inner proteins (MIPs). They also use an RNAi knockdown line to determine the position of DRC11. In principle, I support publication of this work, and I think the data is a valuable addition to the axoneme field. However, there are several major points related to the analysis that must be addressed, in particular related to resolution estimation and exploring structural variation between different DMTs. In addition, there is too much pure speculation about the potential functions of several structures, with no experimental evidence to support these functions. This speculation should be removed or heavily qualified.

We appreciate the positive comments and recognition of the interest and value of the work.

Essential revisions:1) The authors heavily promote the resolution of their structure, claiming "This resolution is the highest cryoET structure yet reported for the 96-nm repeat from any organism". However, there are some serious issues with the resolution estimation, and as a result, I believe the resolution has been overstated.

We have taken this criticism very seriously and have now used the standard Fourier shell correlation calculation server at EMDB to recalculate the resolution of two half maps. The overall resolution reported by the server is 21.8 Å while the resolution of the doublet microtubule with MIPS is 19.0 Å (Figure 3—figure supplement 2). The details are provided below and in the revised text (subsection “3D Structure of the trypanosome 96-nm axonemal repeat”; Subsection “Data processing”). The reviewer is correct that Resmap local resolution provided more optimistic estimation than the standard FSC evaluation. The original Resmap was calculated using the combined map with all particles, which should in theory have better resolution than half maps (i.e., maps from half of the full dataset). Given the recommendations of the reviewer (below), we have also performed local FSCs of specific regions of the structure as requested. The text of the manuscript (subsection “3D Structure of the trypanosome 96-nm axonemal repeat”; Subsection “Data processing”) is revised and Figure 3—figure supplement 2 is made to reflect the updated resolution calculations.

The paper's Abstract claims that the DMT is resolved to 12Å. This is based on the ResMap analysis shown in Supplemental Figure S3. However, this analysis is troubling. First of all, the resolutions on this map appear to primarily range from 12-16Å, whereas the FSC curve shows a global resolution of 21Å at the 0.143 cutoff. If all the local resolutions from ResMap are averaged, the result should be close to 21Å. However, it does not appear that this will be the case, and instead ResMap is estimating resolutions that are at least 5Å better than the FSC. Even the less well-resolved appendage structures that can be seen in this image range from 15-18Å. This is a bit difficult to judge from the figure because the authors have intentionally only shown the backside of the DMT (which reports the highest resolution). They must also show the other side, with all the important accessory structures (similar to Figure 3D), as well as a cross-section slice through the DMT to show the MIPs (similar to Figure 3B). Only then can we see the range of resolutions estimated by ResMap. But even in the view that is shown, the appearance on the DMT is way too speckled, with a huge dynamic range of 12-16Å on the microtubule wall. This noisy surface is a clear sign that something is wrong with the ResMap-the surfaces should look much smoother, with less hotspots of resolution variation.As the ResMap cannot be taken at face-value, a parallel approach should be attempted to estimate local resolution. I recommend using masks to perform two local FSCs – the doublet region and the region containing the appendage structures (OAD, IAD, RS, NRDC). How do these FSCs compare to the global FSC and to the ResMap?

The reviewer has a good point that ResMap typically reports more optimistic resolution numbers. To address this, we have done local FSC calculations for sub‐regions as requested and the resolutions based on these new calculations are included in the revised manuscript subsection “3D Structure of the trypanosome 96-nm axonemal repeat”; Subsection “Data processing” and Figure 3—figure supplement2. We have also provided ResMap views from other angles, as well as a section through the DMT as requested (Figure 3—figure supplement 2).

For the isosurface renderings throughout the paper, the DMT looks properly filtered, but the appendage structures appear to be oversharpened or displayed at a resolution that is too high (their surfaces look noisy). I assume that these maps were uniformly filtered to the same resolution. Was it the global 21Å? I can't seem to find this information in the paper. I would expect that by around 20Å, the holes in the middle of the dynein AAA+ rings would start to become visible, at least as indentations. But in these maps, the AAA+ rings look like round egg-shaped blobs, another sign that the resolution is not as high as claimed.

The reviewer makes good points and we appreciate their evaluation and comments. Our new FSC calculation yields an average resolution of the entire BSF axoneme structure of 21.8 Å based on the 0.143 Fourier shell correlation criterion (Figure 3—figure supplement 2A). The resolutions at different regions vary based on visual inspection, and assessments by both local FSC and *ResMap* (34) calculations (Figure 3—figure supplement 2); the resolution of DMT region with MIPs reached 19.0 Å based on local FSC calculation(Figure 3—figure supplement 2A).The resolution of IAD is 36.1 Å which explains why a hole was not obvious.

We did not sharpen the sub‐tomogram average maps. For rendering, no filters were applied on MIPS but we applied low pass filters on the other components to improve clarity of individual structures described in the text ‐ for the structures in Figure 3C‐E; Figure 4A, B, D; Figure 7A‐E, we filtered the Microtubule, NDRC, RS, IC/LC, OAD and IAD to 30Å and for the structures in Figure 5; Figure 6; Figure 6—figure supplement 1, we filtered the map to 50Å). This information is added to the subsection “3D visualization”MATERIALS AND METHODS.

The authors used a tilt-series acquisition scheme that starts at -60 degrees and thus destroys the high-resolution information before reaching low tilt, as opposed to the much preferred dose-symmetric scheme starting at zero degrees combined with dose-weighting (see the high-resolution HIV work by John Briggs and Wim Hagen). Furthermore, there currently is no way to correct the contrast transfer function (CTF) for low dose cryo-tilt series acquired with the Volta Phase Plate. The authors thus did not perform CTF-correction, meaning that resolution of the average is limited to the first zero of the power spectrum. Given the -0.6 μm target defocus and the large defocus gradient that is present in thick samples such as these axonemes, especially at higher tilts, I anticipate that this first zero would strongly limit the resolution (20-25Å sounds about right, not 12Å). Therefore, I am very cautious of the bold resolution claims made in this paper.

The reviewer’s concern is well taken. We believe this might be because we had omitted details concerning our data processing particularly the part about CTF correction. This has been corrected by adding a detailed paragraph detailing how we performed our reconstruction with CTF correction (Subsection “Data processing” in the revised manuscript). We used in‐focus (~‐0.6um) imaging with VPP phase plate and we always targeted axoneme along the tilt axis. At high tilt angle, the defocus could reach 112nm farther from the targeted ‐0.6um defocus [*i.e.*, from ‐0.49 to ‐0.71um. Note, one of the benefits of using a phase plate is that the CTF is insensitive to the sign of the defocus value being negative (underfocus) or positive (overfocus) (20). We recognized that during imaging, defocus determination is also challenging for VPP data, which might have limited achievable resolution of our results. Indeed, as discussed in responses above, the FSC‐based resolution of our entire 96nm repeat is 21.8 Å with that of the DMT plus MIPS region being 19.0 Å (See Figure 3—figure supplement 2A in the revised manuscript).

Finally, the authors use a "gold-standard" FSC to determine resolution, but it was not clear to me from the methods when exactly the extracted particles were split into two half-sets and averaged independently, as is required for gold-standard assessment. With only 700 total particles, getting two half-sets to 21Å might be challenging. Please explicitly describe how the averaging was performed instead of just "as described previously (75) using PEET".

We have now provided the details of averaging (Subsection “Data processing”) and FSC calculations (Subsection “Data processing”) in the revised manuscript

2) The focus of this paper is to show the evolutionary differences of the T. brucei axoneme. Of course, by far the most distinct feature of trypanosome flagella is the paraflagellar rod (PRF). This structure seems important for axoneme stability under strong forces, a key question the authors sought to address. In Figure 2G, the authors show clear periodic connections between the axoneme and PFR (marked with arrows). It would be very valuable to compare averages of PFR-linked versus non-PRF-linked DMTs. This should be a relatively easy task, just splitting particles into those two categories based on their location with respect to the PFR. Furthermore, the PFR appears to have a fairly regular structure in Figure 2G, so is it possible to generate an average of the PFR itself? Such a structure would be something really new, and would add value to this paper.

As requested, we have now examined individual DMTs and compared PFR‐attached versus not PFR‐attached DMTs – for details, please see the response to Editor comments 1 and 2 above and text of the revised manuscript (subsection “Doublet-specific features of the 96-nm repeat”). Regarding structure of the PFR itself, as discussed in response to Editor comment 2, we feel determining 3D structure of the PFR is beyond the scope of the current paper.

3) The globular density with an 8-nm repeat in Figure 3—figure supplement 3 has a weak signal, and thus appears to have low occupancy in this average. Why is this? Might it have something to do with the connection to the PFR? Judging by Figure 3A, one would expect about a third or a fourth of the DMTs to have a connection to the PFR. The authors should investigate how the occupancy of this density varies between DMTs by producing averages of different DMTs using their radial position relative to the PFR for reference. Without more information, the author's proposed function of this density in regulating dynein binding (subsection “3D Structure of the trypanosome 96-nm axonemal repeat”) is far too speculative.

The density is between protofilaments 7 and 8 of the B‐tubule, so it is not in the correct position to provide contact site for PFR. We agree that the weak signal suggests low occupancy, though we did not resolve a structure here on specific DMTs and thus cannot say whether or not it is specific to a subset of DMTs. Given that this structure is present at or near the site of OAD‐α binding, between protofilaments 7 and 8 in *T. brucei* (Figure 4E), and in *Chlamydomonas* (17), we think it is reasonable to suggest that it may influence dynein binding. We’ve removed speculation regarding impact on IFT motors.

4) Based on the strong signal of the IAD-f IC/LC density, the authors conclude: "An IAD-f IC/LC interdoublet connection is observed between three specific doublet pairs in Chlamydomonas (17). However, the prominence of the IAD-f IC/LC connection in T. brucei suggests it is present between most and perhaps all doublet microtubules, indicating that nexin links in T. brucei include both the NDRC and IAD-f IC/LC. This distinguishes T. brucei axonemes from the known 3D axoneme structures from other organisms (36, 46)." Similar to points 2 and 3 above, if the authors want to make this claim, then they absolutely must examine the density in averages of different DMTs around the T. brucei axoneme.

As requested, we have examined the structure of individual doublet microtubules (see response to Editor comment 1 and 2). The results indicate the f‐connector is present on most doublets. DMT2 is the one exception ‐ it does not have a contiguous connection, although the DMT 2‐3 interface does have a density corresponding to the site of connector attachment on the B‐tubule of DMT 3. This information is now provided in the text (Results section, Figure 6—figure supplement 1Figure).

5) The issues raised in the four points above (problems with resolution estimation and neglecting to analyze structural variation between different DMTs) are all related to the very limited dataset used in this paper. The primary "high resolution" structure in this paper (wild-type BSF) was generated from only 700 particles from 10 tomograms. This is only half a day of acquisition on a Titan Krios microscope (tomograms take about 1 hour each). While I understand it is not eLife policy to ask for more experiments, I think it is completely appropriate in this case for the authors to spend one more day on the microscope with their already prepared cryo-EM samples to acquire 20 more tomograms. This would produce a sufficient dataset to perform classification and look at how specific densities vary between different DMTs (see points 2-4 above). I understand that the Volta Phase Plate used in this study enables averages to be generated with less particles, but I think the 700 particles in the current average are too few to properly do this analysis, and it is not clear to me why the authors chose to proceed with such a limited dataset.

As requested, we have now added additional BSF tomograms to enable analysis of individual DMTs. (For details, see response to Editor comment 1, and reviewer 3 comments 2‐4 above.) For the BSF sample, we obtained a total of 50 tomograms, but selected only the 10 judged as best, based on minimal compression of the axoneme (see Figure 3—figure supplement 1), for sub‐tomogram averaging of all DMTs. As requested by the reviewer, we have now used an additional 24 tomograms (34 in total) to perform sub‐tomogram averaging of individual DMTs. (See Results section, Materials and methods section, Figure 6 and Figure 6—figure supplement 1).

Regarding resolution and number of particles averaged, we point out that the resolution achieved in our work for the averaged BSF 96‐nm repeat (21.8 Å overall resolution at 0.143 FSC criterion, with 763 particles averaged) is within range of structures reported in recent cryoET analyses of axonemes, e.g.:

Stoddard et al., 2018.

Bower et al., 2018.

Lin and Nicastro, 2018.

Dymek et al., 2019.

Owa et al., 2019.

6) The proximal and distal holes in the DMTs look convincing. However, what is not convincing to me is their proposed function of allowing MIPs to be incorporated after completion of the DMT. The holes are tiny, only about as wide as a tubulin monomer (4 nm), and thus do not seem big enough to allow the free transit of MIPs, which are significantly larger than the holes. Perhaps the holes could serve as a location for the start of an "unzipping" event between the A- and B-tubules, which could allow insertion of larger MIP structures such as the ponticulus, but this is completely speculative. I don't think much can be said about the function of the DMT holes at this point.

We have removed speculation on function of holes.

7) The extended discussion of the RingMIP function is highly speculative and should perhaps be omitted or at least down-weighted. Its proposed mechanosensory role is not proven by the data in this paper, just speculated.

We have removed speculation.

8) The authors should be careful with stating the significance of the comparison between developmental stages; there's no clear conclusion from this comparison (other than a low-resolution hint of MIPA3-4). So, putting this in the abstract without indicating the negative result could be considered false advertising. Upon reading the Abstract, I assumed that there were developmental differences, and I was disappointed when I finally discovered at the end of the paper that there were not.

We agree and have removed this comment.

Reviewer #4:Imhof, Zhang et al., present cryo-electron tomography data on preparations of isolated axonemes from the unicellular parasite Trypanosoma brucei, eukaryotic supergroup Excavata. Specifically, they provide a detailed analysis of the 96nm repeat that forms the core structural unit of the axoneme. These data were obtained from two life cycle stages of the organism – the slender bloodstream form which is found in infected mammalian hosts and whose motility is better characterised, and also the procyclic form which is found in the midgut of the tsetse fly vector. Not only are the usual structural features such as the outer dynein arms, radial spokes resolved, but also some fascinating observations of the microtubule inner proteins are provided, which appear to be considerably more abundant in the axonemes of this organism than in others imaged to date.Given that the axoneme is an almost ubiquitous structure that was present in the last eukaryotic common ancestor, the data here are extremely significant and of relevance far beyond the trypanosome and parasitology community. As the authors note, taxonomic sampling of the ultrastructure of the 96nm axonemal repeat is limited, and by providing data from the supergroup Excavata the authors have considerably broadened the perspective onto this structure. The observations of the microtubule inner protein complexes are fascinating and could set the stage for considerable future work.It is important to note that I have no first-hand practical experience of cryo-electron tomography and am therefore not qualified to judge the technical aspects of the manuscript relating to this technique. I am happy for the opinions of the other referee(s) to take precedence on this point.The data appear to be of very high quality and the figures do not require more than cosmetic alteration. No extra practical work seems required, I think, and I have no major concerns. The manuscript could be considerably tightened in order to do full justice to the quality of the results, however.

We appreciate the positive comments and recognition by the reviewer of the significance and relevance of the work to a broad readership.

In particular, the Introduction needs much more detail. The authors should define the 96nm axonemal repeat and explicitly summarise previous work on this structural unit. This is important for placing the results in context and defining the paper's original contribution. The 96nm repeat is currently first mentioned in Results section without any prior introduction. There has been substantial previous work on the morphology and ultrastructure of the trypanosome axoneme, so it is important to emphasise that this work focuses on the structure of the axoneme's 96nm repeat unit specifically. Presently there is considerable potential for being misread, and means that the authors are actually underselling what they have.

The reviewer raises a good point that our focus is the 96‐nm axonemal repeat. We have updated the text throughout to be clear on this. We also explicitly define the 96‐nm repeat in the Introduction based on previous work as well as add a new figure that directly compares the *T. brucei* 96nm repeat to that obtained by cyroET analysis of axonemes from other organisms (Figure 5).

The manuscript also frequently rushes over the results. Panels 2A-2G are not individually cited, nor are 3A-3E, and this trend continues with the other figures. Ensure that all panels are cited, and ideally in figure order (i.e. A-B-C etc) for clarity. Figure 10 is currently not cited at all in the manuscript text. Not all the supplemental figures and movies appear to be cited in the manuscript text – this needs checking. The section on the comparison of the different developmental stages (currently Figure 9) could perhaps be moved so that it comes after Figure 4 instead.

Thank you for pointing out these oversights on our part. We have overhauled the manuscript text to improve clarity and have cited each figure. We have referred to specific figure panels when relevant.

The text would benefit from some proofreading for English, and also for style (e.g. Results should always be presented in past tense, but the present is often used here). Check also that all panel citations are correct – there is some mix-up, particular of Figure 5 and Figure 6. There is also perhaps a bit too much interpretation in the Results section that would fit better in the Discussion section.

The manuscript text has been revised to tighten and to ensure correct in‐text figure references.